

# Boundary description of microstates
# of the two-dimensional black hole

Amr Ahmadain[1★], Alexander Frenkel[1,2†], Krishnendu Ray[3‡] and Ronak M. Soni[1°]

**1** Department of Applied Mathematics and Theoretical Physics, University of Cambridge,
Wilberforce Road, Cambridge, CB3 0WA, United Kingdom
**2** Stanford Institute for Theoretical Physics, Stanford University,
382 Via Pueblo, Stanford CA 94305
**3** Rudolf Peierls Centre for Theoretical Physics, University of Oxford,
Parks Road, Oxford, OX1 3PU, United Kingdom

★ aa2260@cam.ac.uk , † afrenkel@stanford.edu ,
‡ krishnendu.ray@physics.ox.ac.uk , ° rs2194@damtp.cam.ac.uk

## Abstract

We identify the microstates of the non-supersymmetric, asymptotically flat 2$d$ black hole in the dual $c = 1$ matrix quantum mechanics (MQM). We calculate the partition function of the theory using Hamiltonian methods and reproduce one of two conflicting results found by Kazakov and Tseytlin. We find the entropy by counting states and the energy by approximately solving the Schrödinger equation. The dominant contribution to the partition function in the double-scaling limit is a novel bound state that can be considered an explicit dual of the black hole microstates. This bound state is long-lived and evaporates slowly, exactly like a black hole in asymptotically flat space.



# 1  Introduction

The fact that black holes have entropy has been one of the most significant discoveries in the study of quantum gravity. It is a piece of UV physics that we can observe in the IR. In the 40 years since this discovery, however, there are vanishingly few examples in which we understand the microstates that make up this entropy. Examples in which they are understood in the bulk description are extremal maximally supersymmetric black holes — see [1] and its many follow-ups. There are also a larger class of examples where there is some handle in a holographic description, e.g. BTZ black holes (via the Cardy formula) [2], theories that admit quantum mechanical descriptions [3–5], $4d\ \mathcal{N}=4$ SYM [6–8], $3d$ holographic theories [9], etc. Even in this second list, many examples are supersymmetric and extremal.

This paper explores one of the examples from the second list. This is a two-dimensional black hole [10] with no supersymmetry and at finite temperature; however, this black hole is non-standard in that the physics in this background is 'stringy' and its temperature cannot be varied. A lot is known about this solution — this two-dimensional black hole is dual to a one-dimensional matrix quantum mechanics (MQM) system. The finite-temperature partition function of this MQM has been calculated [11, 12] in the path integral formulation, and two candidates for the entropy and energy were found based on this path integral result in [13]. We review the salient features of this model in §2.

In this paper, we aim to go a step further than reproducing the Euclidean partition function in this example — from a mere reproduction of the entropy (a count of the microstates) to a *list* of the microstates. By this, we mean the following. The partition function can of course be reproduced from a trace over a Hilbert space,[1]

$$Z_{\mathrm{MQM}}(\beta) = \operatorname{tr} e^{-\beta H} = \sum_{E \in \operatorname{spec}(H)} e^{-\beta E}. \tag{1}$$

---

[1]It should be noted that the relevant ensemble in our case is grand canonical rather than canonical, and so (1) and (2) are only schematic.

This partition sum, in the thermodynamic limit, is typically dominated by states of approximately equal energy and can be written in the thermodynamic form,

$$Z_{\mathrm{MQM}}(\beta) \approx e^{S(E)} e^{-\beta E} = \mathrm{tr}_{(H=E)} e^{-\beta E} \,, \tag{2}$$

where $S(E)$ is the entropy of the black hole. We have also rewritten this as a trace over a restricted Hilbert space. 'Listing' the microstates of the black hole is tantamount to describing this restricted Hilbert space in a way that does not amount to "all states with the right energy and other charges." There is also a subtlety that these are really microstates of black hole spacetimes rather than black holes themselves; as we will see, we must be careful to disambiguate the microstates of the black hole from states of exterior degrees of freedom. It should be noted that by this standard, only in some extremal supersymmetric cases like [1] (and its follow-ups) have the microstates been listed; in the $\mathcal{N} = 4$ case, this has been done in [14].

One caveat is that the $2d$ black hole is in a phase where the canonical ensemble is not well-defined, and so we have to use a grand-canonical ensemble. In higher-dimensional black holes where the canonical ensemble is well-defined the mass of the black hole is specified by the boundary temperature; in the case we consider, the temperature is fixed but the mass is specified by the chemical potential.

In this paper, we describe this restricted Hilbert space in the MQM system. The MQM has a $PU(N)$ global symmetry, and the restricted Hilbert space turns out to be a certain set of non-singlet irreps of this symmetry, as first pointed out in [15] (see [16–18] for other explicit investigations of the importance of these non-singlet irreps). The set of irreps is determined by the mass of the black hole which, as just mentioned, is related to the chemical potential. As we show in §3, the dimension of this restricted Hilbert space matches one of the two candidates for the entropy found in [13]. However, we also find that this entropy is the count of IR degrees of freedom that decouple from bulk dynamics in the double-scaling limit. We argue that this decoupling can be thought of as a 'renormalisation' in the double-scaling limit in §5.1, and provide a (speculative) pictorial description in §6.3. We thus argue that the actual entropy of the black hole is a value much smaller than found in [13].

To actually describe the microstates of the black hole, we have to find the state of the degrees of freedom that do participate in the bulk dynamics. We analyse the Schrödinger equation to find this state, and find explicitly the state that dominates the grand canonical partition function. It is a novel bound state that we call the 'hole-in-the-world' solution, and we discuss this in §4.2. We argue that the properties of this state *within quantum mechanics* match properties expected from the bulk dual in §6.1.

The plan of the paper is as follows:

1. In §2, we review the duality between the black hole and the MQM system and what is known about the free energy of the black hole.

2. In §3, we show that the degeneracy due to the $PU(N)$ symmetry of the MQM exactly reproduces one of the two candidates for the entropy found in [13].

3. The main section of our paper, §4, reproduces the energetic term in the partition function corresponding to the entropy we find in §3.

   In §4.2, we identify an interesting state, which we call the 'hole-in-the-world' state, with the right energy. We argue that it dominates the fixed-charge partition function in the double-scaling limit in §4.3.

4. The contents of §5 deal with the grand canonical partition function calculated in [11,12]. In §5.1, we address whether the $PU(N)$ symmetry should be gauged or not, and argue that the passage from the ungauged to the gauged quantum mechanics can be thought

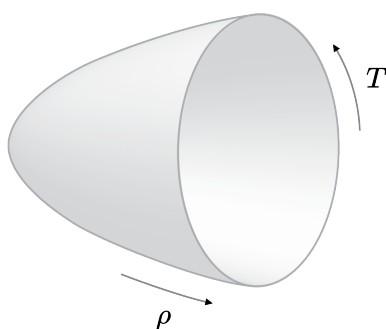

Figure 1: The cigar background.

of as renormalisation in the double-scaling limit. Finally, we comment on the entropy of the black hole in §5.2, and argue that its entropy is much smaller than that found in [13].

5. We conclude with some speculations and discussion in §6. After a broad overview of our results and their limitations, we discuss how the hole-in-the-world state matches expectations about bulk physics in §6.1. §6.2 talks about the implications on the phase structure of the theory. Then, we provide a stringy picture for our analysis in terms of the stretched strings of [16] in §6.3 and discuss a connection between our counting of states and Motzkin walks in §6.4.

**Some notation:** We will use $i, j, k, l$, etc. exclusively to denote elements of $\{1 \ldots N\}$. The imaginary unit is $\iota$.

**Note added:** During the final stages of the preparation of this manuscript, we became aware of the related work [19] which appeared on the arXiv concurrently. Our results, specifically those in §3, have partial overlap with theirs and agree when comparable. Our approaches are complementary.

## 2 Review

The system we work with is the two-dimensional black hole with asymptotically flat boundary conditions [10,20]. It was argued in [11,12] that this is dual to a quantum mechanical system with a single matrix degree of freedom. Some introductions to this set of dualities that we have found useful are [21–27]; a relatively short review is [11]. In this section, we outline the various models in this chain of dualities and summarise the main results of [13].

The cigar is the two-dimensional Euclidean black hole geometry specified by the metric and dilaton,

$$\mathrm{d}s^2 = \mathsf{k}\,\alpha'\,\mathrm{d}\rho^2 + \tanh^2\rho\,\,\mathrm{d}T^2\,, \qquad T \sim T + 2\pi\sqrt{\mathsf{k}\,\alpha'}\,,$$
$$\text{and} \quad \Phi = \Phi_0 - \log\cosh\rho\,. \tag{3}$$

As a manifold, this is the group coset $\frac{SL(2,\mathbb{C})}{SU(2)}\Big/U(1)$. The inverse temperature and mass of this black hole are

$$\beta = 2\pi R \equiv 2\pi\sqrt{\mathsf{k}\,\alpha'}\,, \qquad M = \frac{1}{\sqrt{\mathsf{k}\,\alpha'}}\,e^{-2\Phi_0}\,. \tag{4}$$

Henceforth, apart from some crucial equations, we will set $\alpha' = 1$.

Consider a string propagating on this background,[2] with worldsheet action

$$I_{\text{ws}} = \frac{1}{4\pi\alpha'} \int_\Sigma \mathrm{d}^2\sigma \left[ G_{\mu\nu}(X)\, \partial X^\mu\, \bar\partial X^\nu + \mathscr{R}_\Sigma(X)\, \Phi(X) \right] + I_{\text{WZW}}. \tag{5}$$

Here, $G_{\mu\nu}$ is the metric of the cigar, and $I_{\text{WZW}}$ is a Wess-Zumino-Witten term. $\mathscr{R}_\Sigma$ is the Ricci scalar of the worldsheet. The constant $\mathsf{k}$ is fixed by the requirement that the central charge of this CFT be 26,

$$c_{\text{cigar}} = \frac{3\mathsf{k}}{\mathsf{k}-2} - 1 = 26 \quad \Rightarrow \quad \mathsf{k} = \frac{9}{4}. \tag{6}$$

This means that the $2d$ string theory has a black hole with a fixed temperature given by

$$\beta = 3\pi\sqrt{\alpha'}, \qquad R = \frac{3}{2}\sqrt{\alpha'}. \tag{7}$$

However, its mass can be varied freely.

An important caveat here is that the worldsheet theory is strongly coupled. Since the target space curvature is $\mathscr{O}\left(1/\alpha'\right)$, the $G_{\mu\nu}(X)\partial X^\mu\bar\partial X^\nu$ term is not approximately quadratic in the fields. As a result, the interpretation of this theory as a string theory on a black hole background is not sharp. Calculations are still possible, since the full theory has the properties of a WZW theory. But it is much easier to work in a dual description, which we now describe.

## 2.1 From the cigar to MQM via ER=EPR

This black hole can be described by a matrix quantum mechanics (MQM) through a series of dualities.

1. By FZZ duality [11,12,16], the cigar CFT is dual to the sine-Liouville model.

2. The sine-Liouville model is the zero cosmological constant limit of the sine-Gordon model coupled to Liouville gravity.

3. The sine-Gordon model coupled to Liouville gravity is dual to an MQM with twisted boundary conditions in a double-scaling limit.

The sine-Liouville model is the string theory with worldsheet action

$$I_{\text{sL}} = \frac{1}{4\pi\alpha'} \int_{\tilde\Sigma} \left[ \left(\partial\tilde T\right)^2 + (\partial\phi)^2 + Q\,\mathscr{R}_{\tilde\Sigma}\,\phi + \mathfrak{z}e^{b\phi}\,\cos R\tilde T \right], \qquad \tilde T \sim \tilde T + 2\pi\frac{\alpha'}{R}. \tag{8}$$

The FZZ duality provides an identification between the fields and parameters in the sine-Liouville and cigar CFTs. The fields $\Phi$ and $\phi$ can be identified in the region $\Phi, \phi \gg 1$ (where the background is approximately just flat space with a linear dilaton, i.e. the weak-coupling region). The fields $T$ and $\tilde T$ are $T$-duals of each other; $\mathfrak{z}$ is a chemical potential for momentum $\pm 1$ excitations in the $\tilde T$ direction, and therefore winding $\pm 1$ excitations in the $T$ direction — its value is set by the mass $\mathfrak{z} \propto M^{(2-R)/4}$. The central charge of this theory is [29]

$$c_{\text{sL}} = 2 + 6Q^2. \tag{9}$$

The Liouville-field dressing of any operator in the action with conformal dimension $\Delta$ satisfies

$$b(Q-b) + \Delta = 1. \tag{10}$$

---

[2]It is clear that the cigar and the Lorentzian black hole as metric manifolds are related by an analytic continuation. However, the relation between the string theories defined on the two backgrounds is rather more subtle, as explained for example in [28].



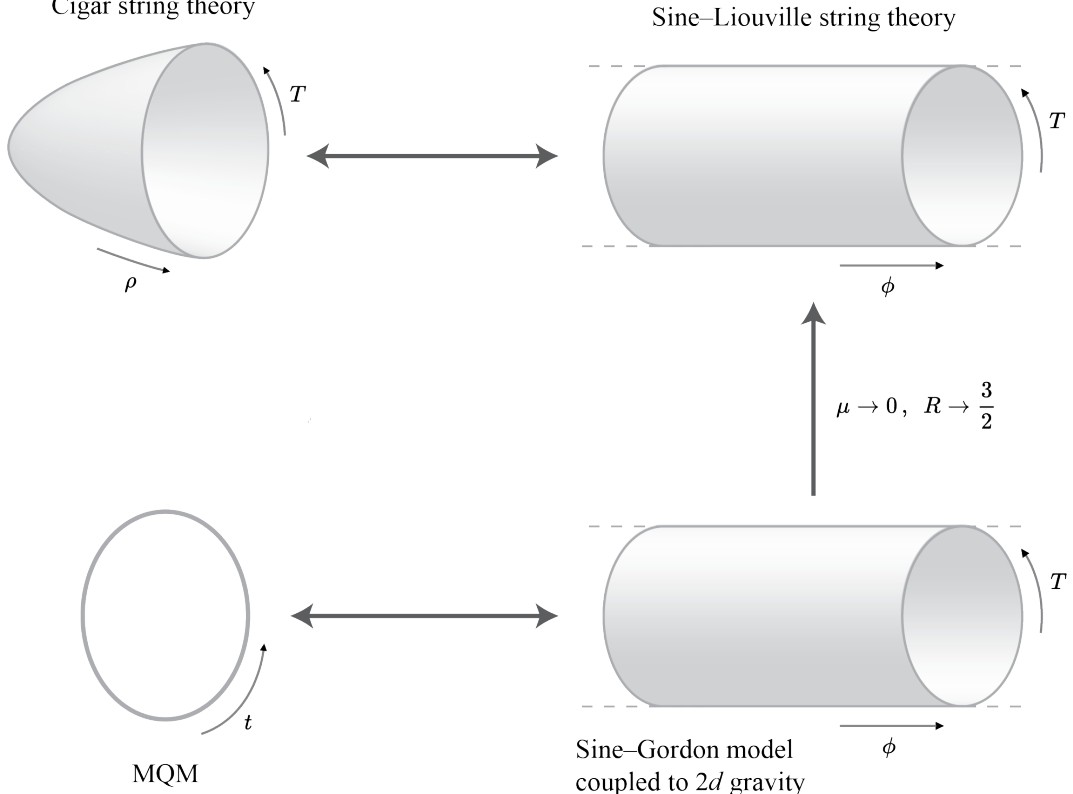

Figure 2: An illustration of the series of dualities that relate the cigar to the MQM.

The undressed sine-Liouville operator $\cos R\tilde{T}$ has conformal dimension $\Delta = R^2$. Matching central charges across the FZZ duality then gives

$$Q = -\frac{1}{b} = \frac{1}{\sqrt{k-2}} \xrightarrow{R=\frac{3}{2}} 2\,. \tag{11}$$

Unlike the cigar, the target manifold in this case is an infinite cylinder, implying an ER = EPR duality [27]. The entanglement is carried by a long string condensate, which is included in the last term in (8). We will see this in greater detail through the course of this work.

An important and non-obvious point about the sine-Liouville theory is that it cannot be studied in perturbation theory in $\mathfrak{z}$. This is because any non-zero $\mathfrak{z}$ can be rescaled to an $\mathcal{O}(1)$ number by shifting the dilaton $\phi$,

$$\mathfrak{z}e^{b(\phi-\phi_0)} = \tilde{\mathfrak{z}}e^{b\phi}\,, \qquad \tilde{\mathfrak{z}} \equiv \mathfrak{z}e^{-b\phi_0}\,. \tag{12}$$

This shift also acts on the $Q\mathcal{R}\phi$ term, and produces a topological term that can be thought of as the string coupling. Thus, the sine-Liouville CFT is not perturbatively related to Liouville theory — this makes it hard to study.

There are two important differences between the sine-Liouville and cigar CFTs. Firstly, the sine-Liouville CFT is on-shell (has $c_{\text{sL}} = 26$) for all $R$ as long as $Q = 2$. Secondly, $k \to 2$ is a strong-coupling limit in the cigar CFT but a weak-coupling limit in the sine-Liouville (since the dressing parameter $b \to 0$ in this limit). Since $k = 9/4$, we henceforth stick to the sine-Liouville side of the duality.

The sine-Gordon model coupled to gravity defines a string theory given by the worldsheet action

$$I_{\text{sG}} = \frac{1}{4\pi\alpha'} \int_{\tilde{\Sigma}} \left[ (\partial T)^2 + (\partial\phi)^2 + Q\mathcal{R}_{\tilde{\Sigma}}\phi + \mu e^{-2\phi} + \mathfrak{z}e^{(R-2)\phi}\cos R\tilde{T} \right]\,. \tag{13}$$

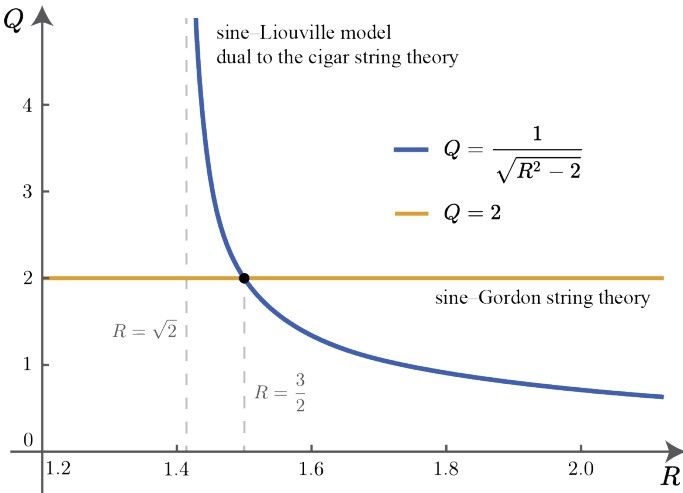

Figure 3: The sine-Liouville theories dual to the cigar and the sine-Gordon theory that can non-anomalously be a worldsheet theory only intersect at $R = 3/2$, when the cigar is on-shell.

Here, we have $T$-dualised again to the $T$ field in the kinetic part of the action, and added a cosmological constant for the Liouville sector. The $\mu \to 0$ limit of this theory is the sine-Liouville model.[3] The $\mathfrak{z} \to 0$ limit of this theory is known as $c = 1$ string theory. Once again, because of the presence of the dilaton, the manner in which we take these perturbative limits is subtle. In sine-Gordon string theory, perturbation theory in $\mathfrak{z}$ is valid where $\mathfrak{z} \ll \mu$, since the zero-mode of $\phi$ can't be shifted while keeping $\mu$ constant. We are interested in the opposite limit, where perturbation theory in $\mu$ is valid.

The sine-Gordon model is dual to a quantum mechanical model with one Hermitian matrix degree of freedom,

$$I_{\mathrm{MQM}} = \int \mathrm{d}t \ \mathrm{tr}\left[\dot{X}^2 - V(X)\right], \qquad V(X) = -\frac{1}{2}X^2 + \frac{g}{N}X^4. \tag{14}$$

Much of the literature uses a potential $V(X) = -X^2/2 + g/\sqrt{N}X^3$; we use this potential following [26], see also [30,31] for the supersymmetric version.

This action has a $PU(N) = U(N)/U(1) = SU(N)/\mathbb{Z}_N$ global symmetry given by

$$X(t) \to U^\dagger X(t) U. \tag{15}$$

Notice that the above transformation is the same for $U$ and $e^{\iota\phi}U$, meaning that the symmetry group is not the unitary group $U(N)$ but the *projective* unitary group $PU(N)$ — as indicated, this is nothing but $U(N)$ quotiented by the overall phase mode. In terms of $SU(N)$, $PU(N)$ is the quotient of $SU(N)$ by its $\mathbb{Z}_N$ centre, which exists because a phase rotation by $e^{\iota j2\pi/N}, j = 0, 1 \ldots N-1$ has determinant 1. As an example, $PU(2) = SU(2)/\mathbb{Z}_2 = SO(3)$.

Some authors, like [16,18,19], gauge this symmetry while others, like [11,32], don't. We will keep this symmetry global for the bulk of this paper; in §5.1 and §6.3, we argue that there is nevertheless a sense in which the gauging 'emerges.'

---

[3]Up to the fact that the dilaton dressing for the vortex term is now $\exp[(R-2)\phi]$ instead of $\exp[-\sqrt{R^2-2}\phi]$. It is easily checked that they agree at $R = 3/2$. The two different expressions correspond to plugging in either $Q = 2$ or $Q = 1/\sqrt{k-2}$ into (10).

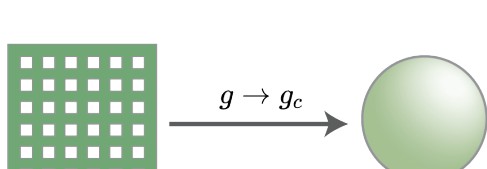

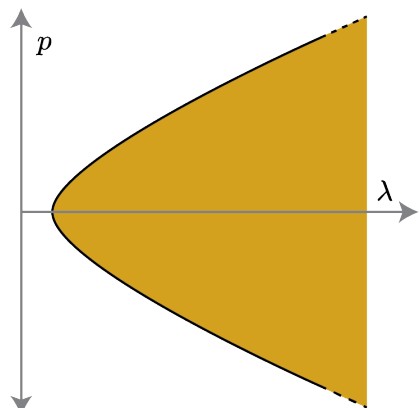

(a) A double-line Feynman diagram appropriate to the MQM becomes a string worldsheet in the double-scaling limit.

(b) The Fermi sea is made by filling up the phase space up to a Fermi level. In the double-scaling limit, the bottom of the well is at $\infty$.

Figure 4: The double-scaling limit.

Precisely, the duality can be written as

$$
e^{-Z_{\text{sG}}(R,\mu,\mathfrak{z})} = \sum_{N=0}^{\infty} e^{2\pi R \mu_F N} \int_{PU(N)} d\Omega \; e^{N \mathfrak{z}_b \, \text{Re}[\text{tr}\,\Omega]} \int_{X(\beta)=\Omega^\dagger X(0)\Omega} e^{-\int_0^\beta dt \; L(X,\dot{X})} \bigg|_{\text{double-scaling limit}} .
\tag{16}
$$

$Z_{\text{sG}}$ is the vacuum partition function of the sine-Gordon model and $e^{-Z_{\text{sG}}}$ is the partition function of the corresponding string field theory. $\mu_F$ is a chemical potential for the size of the matrix, and $\mathfrak{z}_b$ for the non-trivial representations of $PU(N)$. At $\mathfrak{z}_b = 0$, the $\Omega$ integral localises the MQM to the singlet sector of this symmetry — this is dual to $c = 1$ string theory, which is just the sine-Gordon theory at $\mathfrak{z} = 0$. $\mu_F$ is related to the Liouville cosmological constant, $\mu$, in (13), and $\mathfrak{z}_b$ to the fugacity for vortices, $\mathfrak{z}$; the precise relations involve some renormalisations.

Importantly, the duality only emerges in the *double-scaling limit*, which can be thought of as a specific way of taking $N \to \infty$. There are two different but equivalent pictures for this limit, one based on Feynman diagrams and the other based on the Fermi statistics of the eigenvalues of these matrices. We outline the main points of how this is to be understood; more details can be found in the reviews listed at the beginning of this section.

In classic large $N$ fashion, each Feynman diagram can be interpreted as the triangulation of a $2d$ Riemann surface. Since the expansion in the coupling constant $g$ has zero radius of convergence,[4] the series is asymptotic; near a critical value $g_c$ diagrams with $\mathcal{O}\left(\frac{1}{g_c - g}\right)$ vertices dominate the sum. The double-scaling limit is the limit $g \to g_c$, $N \to \infty$, with $N(g_c - g)$ kept finite. In this limit, each diagram becomes a worldsheet and the sum over diagrams becomes a sum over worldsheets; the constant parameter (after renormalisation) becomes the Liouville cosmological constant $\mu$. This is illustrated in figure 4a.

The other equivalent picture for the double-scaling limit is as follows. In the ground state, because of the $PU(N)$ symmetry, we can focus on only the eigenvalues of the matrix $X$. As we explain in §4.1, because of a Vandermonde determinant, these eigenvalues are fermions. In the vacuum, the $N$ fermions fill up the potential up to a Fermi level, see figure 4b. The Fermi level grows monotonically with $g$ up to a value $g_c$ where it becomes equal to the local maximum at 0. We take the $g \to g_c$, $N \to \infty$ keeping the Fermi level at some finite value $-\mu_F \sim \mathcal{O}(N^0)$. This $\mu_F$ is the same as the chemical potential in (16), by standard free fermion arguments. This free fermion theory becomes the dual string *field* theory, with the direction

---

[4]This is most easily seen by considering the case $g < 0$.

given by the log of the eigenvalue being identified with the target space spatial direction $\phi$ (i.e. dilaton), up to a non-local transformation that won't be important to us [25].

The renormalisation is controlled by Khnizhnik-Polyakov-Zamolodchikov (KPZ) [33] dimensions; in particular, the sine-Liouville fugacity and the MQM fugacity are related as

$$\mathfrak{z} = \mathfrak{z}_b \, N^{\frac{2-R}{2}} \,. \tag{17}$$

The relation between $\mu$ and $\mu_F$, which are both renormalised quantities, is a non-universal $\mathcal{O}(1)$ factor. Since we will carry out an MQM calculation, an important consistency check for us will be that, along with this renormalisation, the sine-Liouville answer be independent of $N$.

## 2.2 Free energy and thermodynamics

The matrix partition function can be calculated explicitly [11, 12], and the piece in the free energy corresponding to the genus 0 partition function of the sine-Liouville model for $R < 2$ is

$$\mathcal{F}^{(0)} = -\frac{1}{4}(2-R)^2 \, \mathfrak{z}^{\frac{4}{2-R}} \,. \tag{18}$$

Here, the mass is $M \sim \mathfrak{z}^{\frac{4}{2-R}}$ — it is taken to be large to suppress higher genus effects. On the face of it, this is surprising, since it is a non-zero sphere partition function in string theory. Usually, this vanishes because of the infinite volume of the conformal Killing group, $SL(2,\mathbb{C})$, on the sphere. In the cigar case, however, this result indicates that this volume cancels with the volume of the $SL(2,\mathbb{C})$ factor in the quotient [12].[5]

There are two different approaches to using this result to understand the thermodynamics, as Kazakov and Tseytlin (henceforth, KT) argue [13]. Following their conventions, we will refer to them as the first and second interpretations.

**First interpretation**

1. The MQM free energy is the free energy corresponding to a grand canonical ensemble for a dilaton current. The non-zero value of the free energy to corresponds entirely to the charge term [34] and is proportional to $R - 3/2$.[6]

2. $R$ is varied while keeping the mass constant (i.e. varying $\mathfrak{z}$ as a function of $R$).

This approach gives a free energy,

$$-\beta F_1(R, M) = 2\pi \left( \frac{3}{2} - R \right) M - \frac{R + R^{-1}}{24} \log M \,. \tag{19}$$

The entropy-energy relation is then,

$$S = 3\pi M \,. \tag{20}$$

The vanishing of the leading term in the free energy is attributed to a Hagedorn phase transition at this temperature, that exists for reasons similar to [35].

---

[5]This is not the only way in which the sphere partition function can be non-zero.

[6]Equivalently, we can subtract off the vacuum free energy as in [13].

**Second interpretation**

1. The free energy is given by (18), without any subtractions.

2. The parameter $\mathfrak{z}$ in the sine-Liouville model is interpreted as a chemical potential for vortices. Thus $R$ is varied while keeping $\mathfrak{z}$ fixed, letting $M$ vary as it wills. This is a Gibbons-Hawking-type variation.

Performing an inverse Legendre transform to replace $\mathfrak{z}$ with a vortex number $\Bbbk$, the free energy is

$$-\beta F_2(R, \Bbbk) = \left(2 - \frac{\beta}{2\pi}\right)\Bbbk \log \frac{\Lambda \alpha'}{\sqrt{\Bbbk}} + \mathcal{O}(\Bbbk). \tag{21}$$

Here, $\Lambda$ is a UV cutoff on the worldsheet with dimensions of inverse length, and $\Bbbk$ is related to $\mathfrak{z}$ and the mass (at leading order) as

$$\Bbbk \approx \mathfrak{z}^{\frac{4}{2-R}} \propto M. \tag{22}$$

This is finite but large in the sine-Liouville theory, and so we have $1 \ll \Bbbk \ll N$ in the dual MQM. For the energy and entropy (setting $\alpha' = 1$ again), this gives,

$$M = \frac{1}{2\pi}\Bbbk \log \frac{\Lambda}{\sqrt{\Bbbk}}, \qquad S = 4\pi M. \tag{23}$$

We also note that we can identify $\Lambda \alpha' \sim N$ based on the arguments of [36,37].[7]

The authors of [13] concluded with the remark that the best way to decide between the two interpretations would be to perform a microscopic calculation in the MQM. Note that the difference between the two interpretations is not just a matter of convention or ensemble; the relation between entropy and energy is a statement about the Hilbert space that is either true or not, and we can check which one agrees with the Hilbert space structure. This is what we set out to do — we find that the Hamiltonian analysis suggests the second interpretation is correct.

## 3 Entropy in the ungauged model

In this section, we reproduce the entropy in the KT result (21) by assuming that the $PU(N)$ symmetry is a global symmetry rather than a gauge symmetry. This entropy was originally calculated in [38], but we clarify some points and fill in some details. We present two calculations, one in §3.1 and the other in §3.2, and find that they agree. The calculation in §3.1 is quick and direct but badly motivated; the calculation presented in §3.2, which is a version of the calculation in [38], is slightly longer but better motivated. A novel calculation of entropy in a phase with $\Bbbk \gg N^2$ is presented in Appendix A; we have not studied the Hamiltonian in this phase.

The entropy we calculate in the ungauged model reproduces the proposed black hole entropy Kazakov and Tseytlin identify in [13]. We find that the entropy is entirely due to the large degeneracy dictated by the $PU(N)$ symmetry, and not due to a large number of interesting interacting degrees of freedom. We will therefore propose an interpretation in §5.2 and §6.3 that it is better to think of the KT entropy not counting the black hole degrees of freedom, but of some IR degrees of freedom that decouple in the double-scaling limit. In particular, we will argue in §5.1 that in the double-scaling limit this symmetry is effectively gauged and the

---

[7] [36,37] find an expression very similar to this for the free energy, with $\Lambda$ being a worldsheet cutoff. In this case, the worldsheet cutoff is controlled by $N$.

large entropy we calculate here is therefore not relevant to bulk physics. An important intermediate result in this section that retains its importance in the gauged model is the fact that a single irrep, illustrated in figure 5, dominates the subspace with a large number of adjoints. This result has also been found in [19].

## 3.1 Direct calculation

The direct calculation proceeds based on the simple observation that $\Bbbk$ is the charge conjugate to the fugacity for singlets and non-singlets, $\mathfrak{z}_b$. Since the basic representation of the symmetry in the model is an adjoint representation (15), the subspace with fixed $\Bbbk$ is the one constructed out of $\Bbbk$ copies of the adjoint representation. The dimension of this reducible representation, remembering to quotient by the symmetric group $\mathsf{S}_{\Bbbk}$ because the adjoints are indistinguishable, is

$$\log \dim \left( \mathscr{H}_{\text{adj}}^{\otimes \Bbbk} / \mathsf{S}_{\Bbbk} \right) \approx \log \frac{N^{2\Bbbk}}{\Bbbk!} \approx 2\Bbbk \log \frac{N}{\sqrt{\Bbbk}} \, . \tag{24}$$

This is exactly the entropy in (21).

## 3.2 Calculation using irreps

While we seem to have reproduced the KT entropy, we have no reason to believe that the subspace whose dimension we just counted above is even approximately degenerate. Exact degeneracy is, however, guaranteed within each irreducible representation of the $PU(N)$ group. We now proceed break up the fixed $\Bbbk$ subspace into irreps of $PU(N)$ and compute their dimensions — we will see that we find the same answer. Further, we show that the mistake in [38] that was pointed out by [32] is subleading. Substantial parts of this section are a re-iteration of the work in [38], as well as [32,39], so that the calculation may be presented as a unified whole.

The irreps of $U(N)$ are usually given by Young diagrams (YDs) with at most $N$ rows. Each box corresponds to a copy of the fundamental representation $\mathscr{H}_f$; boxes in the same row are symmetrised and those in the same column are anti-symmetrised. We modify this prescription, following [38, 40, 41], to also allow for 'anti-boxes,' which correspond to copies of the anti-fundamental representation $\mathscr{H}_{\bar{f}}$. An example irrep, using green to denote the anti-boxes, is

$$r_{\text{eg}} = \tag{25}$$

With these anti-boxes included, the conjugate representation is constructed by mirroring the diagram along the horizontal axis. We can ensure that we are not over- or under-counting by restricting the maximum number of rows to $N/2$.[8]

The irreps of $PU(N)$ are then given by Young diagrams with an *equal* number of boxes and anti-boxes. This equality ensures that the overall phase mode of $U(N)$ acts trivially in this irrep. So, for example, (25) has 3 extra anti-boxes compared to boxes — meaning that it has charge $(-3)$ under the phase mode. A valid irrep of $PU(N)$ is

$$r_{\text{eg}}^{PU(N)} = \tag{26}$$

---

[8]More precisely, we have to restrict the sum of the number of rows and the number of anti-rows to be at most $N$. The precise specification then depends on whether $N$ is even or odd in an obvious way.

As pointed out by [32], the authors of [38] erroneously imposed the stronger condition that the irrep be self-conjugate, i.e. that the YD be symmetric under reflection.

Once again, the group $PU(2) = SU(2)/\mathbb{Z}_2$ makes for a simple example. In this case, the fundamental is its own conjugate and so anti-boxes and boxes are identical. That there are an equal number of boxes and anti-boxes is simply the requirement that there are an even number of boxes; these are just the integer spin representations of $SU(2)$, which are well known to be the representations of $SO(3)$.

States in the irrep are given by *Young tableaux* (YTs) — this is a filling in of the boxes with numbers taken from $\{1, \ldots, N\}$ following a set of rules detailed in [40, 41]. Thus,

$$\mathscr{H}_{r_{\text{eg}}} = \text{span}\left\{ \left| \begin{array}{ccccccc} j_3 & j_2 & j_1 & i_1 & i_2 & i_3 & i_4 \\ & j_5 & j_4 & i_5 & i_6 & & \\ & & j_6 & & & & \end{array} \right\rangle \middle| j_a, i_a \in \{1, \ldots N\} \text{ following the rules below.} \right\}. \tag{27}$$

The number in the (anti-)box denotes the corresponding basis element in the (anti-) fundamental representation. The rules can be found in [40, 41]; they follow more or less directly from the fact that two boxes in the same row are symmetrised and two boxes in the same column are anti-symmetrised, along with the requirement that every basis element in the irrep have only one associated YT. The two most important rules are

- $R_1$: The entries in both boxes and anti-boxes increase from top to bottom.

- $R_2$: The entries for boxes (anti-boxes) increase from right to left (left to right).

These follow from the fact that different YTs, e.g. $\boxed{1\ \ 2}$ and $\boxed{2\ \ 1}$, would represent the same state due to the (anti-)symmetrisation rules. Note that these two rules apply separately to the boxes and the anti-boxes.

The new complication due to the presence of both boxes as well as anti-boxes is as follows. The state

$$\sum_{i=1}^{N} \left| \boxed{i\ \ i} \right\rangle \in \text{singlet}. \tag{28}$$

Any fundamental and any anti-fundamental can pair up into such a 'trace.' To define a true irrep then, we need to project out such possibilities; [40, 41] use a third rule stating that "If $r(i)$ and $r(\bar{i})$ are the lowest rows containing the indices $i$ and $\bar{i}$, $r(i) + r(\bar{i}) \leq i$" to deal with this subtlety; they exclude enough possible YTs that a trace is never possible but keep enough so that the irrep space has the right dimension. We will not impose this rule, and deal with the possibility of traces explicitly, aided by the fact that we are working in the limit of large $N$.

With all of this in place, let us return to the tensor product of $\mathbb{k}$ adjoints that we considered in the previous section — this can be broken up into a direct sum of irreps of $PU(N)$ with at most $\mathbb{k}$ boxes and $\mathbb{k}$ anti-boxes. For the moment, let us focus on the YDs with exactly $\mathbb{k}$ boxes and anti-boxes. In the large $N$ limit, each of these irreps, in turn, can be written as a product of an irrep $r_1$ with $\mathbb{k}$ boxes, and $\bar{r}_2$ with $\mathbb{k}$ anti-boxes. Let us first calculate the dimension, i.e. the number of YTs, corresponding to the YD $r_1$. A useful trick to do this counting follows from a simple observation [38]: since the number of potential labels is much larger than the number of boxes, repetitions are non-generic. Thus the number of ways to choose the labels is just $\binom{N}{\mathbb{k}}$. Such a Young tableau, constructed out of numbers with a total ordering, is called a *standard Young tableau* (sYT) and labels a state in an irrep of the symmetric group $\mathsf{S}_{\mathbb{k}}$. Letting $d_{r_1}^{\mathsf{S}_{\mathbb{k}}}$ denote the dimension of this irrep of $\mathsf{S}_{\mathbb{k}}$, we have

$$\dim \mathscr{H}_{r_1} = \binom{N}{\mathbb{k}} d_{r_1}^{\mathsf{S}_{\mathbb{k}}}. \tag{29}$$

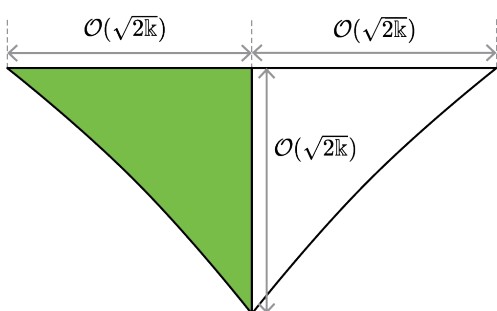

Figure 5: The shape of the YD that dominates the counting of states in the $\Bbbk \ll N$ phase.

Of course, the dimension $d_{r_1}^{S_\Bbbk}$ depends on the particular shape of the irrep of $S_\Bbbk$. An explicit verification of this trick can be found in [42]. Using the same argument for the anti-boxes, we find

$$\dim \mathscr{H}_{r_1 \otimes \bar{r}_2} = \binom{N}{\Bbbk}^2 d_{r_1}^{S_\Bbbk} d_{\bar{r}_2}^{S_\Bbbk}. \tag{30}$$

Summing over all such irreps with $\Bbbk$ boxes, we find

$$\sum_{r_1, \bar{r}_2 | \Bbbk} \dim \mathscr{H}_{r_1 \otimes \bar{r}_2} = \binom{N}{\Bbbk}^2 \left( \sum_{r | \Bbbk} d_r^{S_\Bbbk} \right)^2. \tag{31}$$

The sum is a standard result in combinatorics (e.g. see Theorem 8.26b in [43]) and the result is called the $\Bbbk^{th}$ telephone number. At large $\Bbbk$, it is approximated by [44]

$$\sum_{r | \Bbbk} d_r^{S_\Bbbk} = \sqrt{\Bbbk!}. \tag{32}$$

The sum of dimensions of all $PU(N)$ irreps with $\Bbbk$ boxes and $\Bbbk$ anti-boxes is thus

$$\binom{N}{\Bbbk}^2 \Bbbk! \approx \left( \frac{N}{\sqrt{\Bbbk}} \right)^{2\Bbbk}, \tag{33}$$

reproducing again the result of §3.1.

Of course, it is not correct to sum over all irreps — degeneracy is only guaranteed within an irrep of $PU(N)$, whereas we have counted the dimension of a *reducible* representation. This is in fact the same unjustified step we took in §3.1. However, let us note something simple:

$$\left( \sum_{r | \Bbbk} d_r^{S_\Bbbk} \right)^2 \approx \Bbbk! = |S_\Bbbk| = \sum_{r | \Bbbk} \left( d_r^{S_\Bbbk} \right)^2. \tag{34}$$

In other words, at large $\Bbbk$, the dimension of the irrep of $S_\Bbbk$ has zero variance, indicating that the sum is dominated by a single YD. This is an old result [45]; they found that the dominant irrep is roughly a right isosceles triangle with somewhat concave sides i.e., the longest column, row and anti-row have approximately $\sqrt{2\Bbbk}$ boxes each. This irrep is shown schematically in figure 5. This domination by a single diagram is why we were able to sum over all irreps and still reproduce the entropy of (21). Another interesting consequence is that our counting is actually valid in the regime $1 \ll \Bbbk \ll N^2$ and not just $1 \ll \Bbbk \ll N$ as originally claimed.

Finally, let us deal with the possibility of traces; naively, the count above includes states in smaller irreps since we didn't project out traces. Since a trace causes the number of (anti-)

boxes in the YD to reduce by one, the counting above shows that the number of these states is approximately $(N/\sqrt{\Bbbk})^{2\Bbbk-2}$ which is a factor of $N^2$ smaller. Further, the leading corrections in the count just given (from corrections to Stirling's formula and the approximation we made for the telephone number) is a factor of $e^{\#\Bbbk}$, which is a correction of $\mathcal{O}(\Bbbk)$ in the entropy. Thus, the uncertainty from the shape of the YD (comparable to the correction in the telephone number) dominates the uncertainty from the possibility of traces.

Since both the boxes and anti-boxes are dominated by the same irrep, the dominant irrep in the full counting is self-conjugate. This is why our result agrees with the counting in [38] even though they restricted their count to only self-conjugate representations. It is interesting to note that in a very different context, the authors of [46] also found that the microstates of an incipient black hole are related to YDs that are almost an isosceles triangle.

# 4 Energy

Having found that the dominant degenerate space with $\Bbbk$ boxes reproduces the entropy of KT, we now calculate its energy. In §4.1, we decompose the Hilbert space in a useful way and set up the Schrodinger equation problem that we will attempt to solve. To reproduce the fixed-charge partition function (21), we proceed in two steps. First, in §4.2, assuming that within a fixed $\Bbbk$ sector the canonical and microcanonical ensembles are equivalent, we construct a state whose energy matches the KT energy; the construction of this state is the central result of our paper. In §4.3, we justify this assumption, arguing that the two ensembles are indeed equivalent within the sector.

Section 4.2.3 contains the central new result of our paper: the construction of a state when $1 \ll \Bbbk \ll N$ in the irrep that dominates the counting above whose energy matches that from (21) on the nose. This state is a new, interesting bound state that is reminiscent of a black hole. We explore this facet, and argue that the dominance of such a bound state is exactly what is needed for the MQM to be dual to the sine-Liouville string theory, in §6.1.

## 4.1 Decomposition of the Hilbert space

The Hilbert space of the MQM is

$$\mathcal{H}_{\mathrm{MQM}} = \mathrm{span}\left\{ |X\rangle \,\Big|\, X^\dagger = X \right\}. \tag{35}$$

We will first decompose this Hilbert space into its eigenvalue and angular directions and then "Fourier transform" the angular directions using the Peter-Weyl theorem. This has the nice benefit of isolating the action of the $PU(N)$ symmetry.

We decompose the matrix $X$ as

$$X(t) = U^\dagger(t)\Lambda(t)U(t), \quad \text{s.t.} \quad \Lambda_{ij} = \lambda_i \delta_{ij}, \ \lambda_i \in \mathbb{R}, \ i > j \Rightarrow \lambda_i \geq \lambda_j, \quad U \in U(1)^N \backslash U(N). \tag{36}$$

The conditions on the diagonal and unitary matrices are required to make the decomposition uniquely defined — we explain each of them in turn. First, the eigenvalues need to be ordered since we can always rearrange them without changing $X$ by absorbing an element $P_\sigma \in \mathsf{S}_N \in U(N)$ into $U$,

$$U^\dagger \Lambda U = (P_\sigma U)^\dagger (P_\sigma \Lambda P_\sigma^\dagger)(P_\sigma U). \tag{37}$$

Thus, denoting the space of $N \times N$ diagonal matrices by $D(N)$, we require $\Lambda \in D(N)/\mathsf{S}_N$.[9] Similarly, the $U(N)$ needs to be *left*-quotiented by $U(1)^N$ to account for the gauge symmetry,

$$X = U^\dagger \Lambda U = U^\dagger e^{-\iota\Theta} \Lambda e^{\iota\Theta} U, \qquad \Theta_{ij} = \theta_i \delta_{ij}. \tag{38}$$

---

[9]As we will clarify below, $D(N)/\mathsf{S}_N$ is not a strictly correct description of the space of $\Lambda$s.

The global symmetry (15) is given by a *right*-action on $U$, $U \to UV$. There is also a left-action, $U \to VU$, which doesn't commute with the Hamiltonian. The measure over the space decomposes into

$$\mathrm{d}^{N \times N} X = \frac{\mathrm{d}^N \lambda \, \mathrm{d}U}{N! \left[ \mathrm{vol}\, U(1) \right]^N} \Delta(\lambda)^2, \qquad \Delta(\lambda) = \prod_{i<j} \left( \lambda_i - \lambda_j \right). \tag{39}$$

The two terms in the denominator are the size of $\mathsf{S}_N$ and the volume of $U(1)^N$ in the Haar measure; we will henceforth absorb them into normalisation factors. $\Delta(\lambda)$ is the Vandermonde determinant, whose square is the Jacobian for this change of basis. Thus, finally, we have

$$\mathscr{H}_{\mathrm{MQM}} = \mathscr{H}_{D(N)/\mathsf{S}_N} \bigotimes \mathscr{H}_{U(1)^N \backslash U(N)}, \tag{40}$$

where the two factors are spaces of normalisable functions over the respective spaces.

$\mathscr{H}_{D(N)/\mathsf{S}_N}$ can be thought of as the space of wavefunctions on the subspace of $\mathbb{R}^N$ satisfying the ordering conditions. To extend it to all of $\mathbb{R}^N$, we need to contend with the fact that permutation of eigenvalues also requires absorbing a left-action into the unitary, meaning that the extension is therefore not independent of the state in the angular directions. However, since we are interested in the case $1 \ll \Bbbk \ll N$, we can forget about this subtlety and extend it to $\mathbb{R}^N$ as a symmetric wavefunction [17]. Furthermore, because the inner product on these wavefunctions is

$$\langle \psi_1 | \psi_2 \rangle = \int \mathrm{d}^N \lambda \, \Delta(\lambda)^2 \, \psi_1^*(\lambda) \, \psi_2(\lambda), \tag{41}$$

it is conventional to absorb a factor of $\Delta(\lambda)$ into the wavefunction to get completely *anti*-symmetric wavefunctions (ignoring the angular directions) and a conventional measure for the inner product.

We will largely be concerned with the angular part of the Hilbert space, $\mathscr{H}_{U(1)^N \backslash U(N)}$. This Hilbert space can be thought of as simply $\mathscr{H}_{U(N)}$ specified by a gauge constraint. To do so, we first parametrise the algebra $\mathfrak{u}(N)$ in the weight basis $\left\{ H_i, T_{ij}, \tilde{T}_{ij} \mid i,j \in \{1, \dots N\}, \ i<j \right\}$. In the fundamental representation, these basis elements have the matrix representations

$$\begin{aligned}
(H_i)_{kl} &= \delta_{kl} \, \delta_{ki}, \\
\left( T_{ij} \right)_{kl} &= \delta_{ki} \, \delta_{lj} + \delta_{li} \, \delta_{kj}, \\
\left( \tilde{T}_{ij} \right)_{kl} &= -\iota \left( \delta_{ki} \, \delta_{lj} - \delta_{li} \, \delta_{kj} \right).
\end{aligned} \tag{42}$$

The generators $T_{ij}$ and $\tilde{T}_{ij}$ can respectively be thought of as the Pauli matrices $\sigma_1$ and $\sigma_2$ acting on the two-dimensional subspace corresponding to $i,j$. To each of these generators correspond two operators on $\mathscr{H}_{U(N)}$, the left and right actions,

$$\begin{aligned}
e^{\iota \left[ \alpha^{ij} \hat{T}_{ij}^L + \beta^{ij} \hat{\tilde{T}}_{ij}^L + \gamma^i \hat{H}_i^L \right]} |U\rangle &= \left| e^{\iota \left[ \alpha^{ij} T_{ij} + \beta^{ij} \tilde{T}_{ij} + \gamma^i H_i \right]} U \right\rangle, \\
e^{\iota \left[ \alpha^{ij} \hat{T}_{ij}^R + \beta^{ij} \hat{\tilde{T}}_{ij}^R + \gamma^i \hat{H}_i^R \right]} |U\rangle &= \left| U e^{-\iota \left[ \alpha^{ij} T_{ij} + \beta^{ij} \tilde{T}_{ij} + \gamma^i H_i \right]} \right\rangle.
\end{aligned} \tag{43}$$

Now, the subspace $H_{U(1)^N \backslash U(N)}$ is given by

$$\begin{aligned}
\mathscr{H}_{U(1)^N \backslash U(N)} &= \left\{ |v\rangle = \int \mathrm{d}U \, v(U) |U\rangle \ \middle| \ v\left( e^{\iota \Theta} U \right) = v(U) \right\} \\
&= \left\{ |v\rangle \in \mathscr{H}_{U(N)} \ \middle| \ \hat{H}_i^L |v\rangle = 0 \right\}.
\end{aligned} \tag{44}$$

The second equation arises from the fact that the change $U \to e^{\iota \Theta} U$ is implemented by the exponential of $\Theta^i \hat{H}_i$. We will refer to the gauge constraint as the *zero-weight condition*.

To implement this constraint, it is useful to use the Peter-Weyl theorem to go to the "Fourier transformed" basis

$$|r,\alpha,\beta\rangle \equiv \sqrt{d_r} \int dU\, D^r_{\beta\alpha}(U^\dagger)|U\rangle\,,$$

$$|U\rangle = \sum_r \sum_{\alpha,\beta} \sqrt{d_r}\, D^r_{\alpha\beta}(U)\,|r,\alpha,\beta\rangle\,. \tag{45}$$

Here, $r$ is an irrep of $U(N)$, usually represented by a YD. $\alpha,\beta$ are two different, unrelated YTs filling in the YD corresponding to $r$.[10] In the $N=2$ case, with standard $SU(2)$ notation, the states are $|j,m,m'\rangle$ and the coefficients are matrix elements of Wigner $D$-matrices. The change of basis is unitary because of the orthogonality relations

$$\int dU\, D^r_{\beta\alpha}(U^\dagger)D^{r'}_{\gamma\delta}(U) = \frac{1}{d_r}\,\delta^{rr'}\,\delta_{\alpha\gamma}\delta_{\beta\delta}\,, \qquad \sum_r\sum_{\alpha,\beta} d_r\, D^r_{\beta\alpha}(U^\dagger)D^r_{\alpha\beta}(U') = \delta(U^{-1}U')\,. \tag{46}$$

The most useful fact about these states is that they 'decouple' the right and left actions; denoting by $\hat{V}^L,\hat{V}^R$ the left and right multiplications by $V$ respectively, we have

$$\hat{V}^L\,|r,\alpha,\beta\rangle = |r,\gamma,\beta\rangle\, D^r_{\gamma\alpha}(V)\,,$$

$$\hat{V}^R\,|r,\alpha,\beta\rangle = D^r_{\beta\delta}(V^\dagger)\,|r,\alpha,\delta\rangle\,. \tag{47}$$

Indeed, we see that the left and right actions act on the two indices (i.e. YTs) separately. We remind the reader that the right action is a symmetry of the theory, and therefore the YTs denoted by the $\beta$ index here are the ones we counted in §3 (more correctly, in that case $r$ was an irrep of $PU(N)$ — a restriction that we will impose presently).

In the Fourier-transformed basis, the zero-weight condition is

$$\hat{H}^L_i\,|r,\alpha,\beta\rangle = |r,\gamma,\beta\rangle\, D^r_{\gamma\alpha}(H_i) = 0\,. \tag{48}$$

We can simultaneously diagonalise all the weights in $\mathscr{H}_r$, and then just pick the zero-weight subspace $\mathscr{H}^{(0)}_r$ as the span of all $\alpha^{(0)}$ that have all weights vanishing. Thus, we have

$$\mathscr{H}_{U(1)^N\backslash U(N)} = \bigoplus_r \mathscr{H}^{(0)}_r \otimes \mathscr{H}_{\bar{r}} = \text{span}\left\{\left|r,\alpha^{(0)},\beta\right\rangle\right\}\,. \tag{49}$$

We emphasise that the zero-weight condition is a condition on the irreps of $U(N)$ as well as the states within the representations. In particular, there are irreps of $U(N)$, like the fundamental representation, that don't contain any zero-weight states.

The set of irreps that contain zero-weight states are exactly the irreps of $PU(N)$. We can see this as follows: $H_i$ annihilates every basis vector of the fundamental (anti-fundamental) except the $i^{th}$ one where it takes the value $1(-1)$. Thus, the zero-weight condition is simply [40,41]

$$\forall i,\quad \hat{H}^L_i\,|\text{YT}\rangle = \left\{\#\boxed{i} - \#\boxed{\phantom{i}}\!\!\!\color{green}{i}\right\}|\text{YT}\rangle = 0\,. \tag{50}$$

This cannot vanish for all $i$ unless the number of boxes and anti-boxes agree in the YD, as was pointed out in [32]. This is an alternative way of deriving the restriction to irreps of $PU(N)$. This condition is actually stronger, and says that the boxes and the anti-boxes have to carry the same set of labels.

---

[10]$\beta$ should actually be considered a basis element in the conjugate irrep, but there is a canonical map and so it can also be considered a YT of the same shape.

To calculate the energy, we first decompose the Hamiltonian according to (36). Plugging in the decomposition (36) of $X$ into the action (14), we find

$$I_{\text{MQM}} = \int dt \; \text{tr}\left[\frac{1}{2}\dot{\Lambda}^2 - V(\Lambda) + \frac{1}{2}\left[\dot{U}U^{\dagger}, \Lambda\right]^2\right]. \tag{51}$$

Notice that the action contains $\dot{U}U^{\dagger}$ but not $U^{\dagger}\dot{U}$. The operator $\dot{U}U^{\dagger}$ generates a change from the left — $\left(1 + dt\,\dot{U}U^{\dagger}\right)U = U + dt\,\dot{U}$ — whereas $U^{\dagger}\dot{U}$ generates a change from the right; the fact that the latter doesn't appear in the action but the former does is indicative of the fact that only the latter is a symmetry. The Hamiltonian is

$$H = \sum_i\left[-\frac{1}{2\Delta(\lambda)}\frac{\partial^2}{\partial\lambda_i^2}\Delta(\lambda) + V(\lambda_i)\right] + \frac{1}{4}\sum_{i<j}P_{0\text{wt}}\frac{\hat{T}_{L,ij}^2 + \hat{\tilde{T}}_{L,ij}^2}{(\lambda_i - \lambda_j)^2}P_{0\text{wt}}. \tag{52}$$

Once again there are only left-action generators in the Hamiltonian, and the weight operators don't appear at all, since the weights are fixed by the gauge condition. $P_{0\text{wt}}$ is an explicit projector onto the zero-weight subspace; this is necessary because $\hat{T}_{L,ij}^2$ doesn't commute with it in arbitrary representations. We will drop the $L$ subscript henceforth since the right-acting operators don't play a role at all. We will also drop the hats since only the quantum operators appear in the following discussion. In what follows, we will find it useful to define

$$H_{\text{ns}} \equiv \sum_{i<j}H_{ij}, \qquad H_{ij} \equiv \frac{1}{4}\frac{T_{ij}^2 + \tilde{T}_{ij}^2}{\left(\lambda_i - \lambda_j\right)^2}, \tag{53}$$

where $H_{\text{ns}}$ is the Hamiltonian of the *non-singlet* sector. It has precisely the form of a spin-Calogero model (see [47] for a discussion in the context of MQM), an integrable system that has been extensively studied in condensed matter, MQM, and mathematics literature.

The singlet sector, which is annihilated by $H_{\text{ns}}$, is just $N$ non-interacting fermions. Denoting the ground state of this sector by $|0\rangle_{\text{sing}}$, its energy is is obtained by filling up $N$ levels of a Fermi sea of eigenvalues,[11]

$$E_0 \sim \mathcal{O}(N^2). \tag{54}$$

In comparison, $H_{\text{ns}}$ scales with $\Bbbk \ll N^2$ and can thus be treated as a perturbation [39]. With $H_{\text{ns}}$ turned off, the Hamiltonian does not act on the angular directions at all, and there is a huge degenerate subspace, given (after dropping the right-action index) by

$$\mathcal{H}_{\text{degen}} = |0\rangle_{\text{sing}} \otimes \text{span}\left\{\left|r, \alpha^{(0)}\right\rangle\right\}. \tag{55}$$

$H_{\text{ns}}$ breaks this degeneracy, and we have to use the techniques of degenerate perturbation theory. The perturbed energy levels are the eigenvalues of the operator $P_{\text{degen}}H_{\text{ns}}P_{\text{degen}}$, where $P_{\text{degen}}$ is the projector onto the degenerate subspace (55). Below, we will find that this will not be enough to reproduce the KT partition function — the non-singlets will backreact on the eigenvalue distribution in the dominant state. This is nonetheless a useful starting point.

The eigenvalue ground state is characterised by a phase space density function

$$\rho(\lambda, p_\lambda) = \frac{1}{2\pi}\,\theta\left(E_F - \frac{p_\lambda^2}{2} - V(\lambda)\right). \tag{56}$$

Integrating over momenta gives the eigenvalue density function

$$\rho(\lambda) = \frac{1}{\pi}\sqrt{2E_F - 2V(\lambda)} \approx \frac{1}{\pi}\sqrt{-2\mu_F + \lambda^2} \xrightarrow{\lambda \gg \sqrt{2\mu_F}} \frac{\lambda}{\pi}, \tag{57}$$

---

[11]We subtract off this zero-point energy in the double-scaling limit.

where $E_F = -\mu_F$ is the Fermi energy we mentioned in §2; the double scaling limit of interest involves keeping $\mu_F \sim \mathcal{O}(1)$ as $N \to \infty$. Here, we have replaced the quartic part of the potential by a wall at $\lambda_c \sim \mathcal{O}(\sqrt{N})$, since the precise form of the cutoff doesn't matter in the large $N$ limit. The number of eigenvalues between $\lambda_1$ and $\lambda_2$ is $\int_{\lambda_1}^{\lambda_2} \rho(\lambda)\,\mathrm{d}\lambda$ — the eigenvalue density is then normalised as[12]

$$\int_{\sqrt{2\mu_F}}^{\lambda_c} \rho(\lambda)\,\mathrm{d}\lambda = N\,. \tag{58}$$

We will for many purposes drop the $\mu_F$-dependence in what follows, reinstating it only when necessary. Any expectation value in the ground state $|0\rangle_{\text{sing}}$ is controlled by this eigenvalue density — thus, the projector $P_{\text{degen}}$ ensures that the part of $H_{\text{ns}}$ acting on the eigenvalues is determined by this eigenvalue density.

Clearly then, the eigenvectors of $P_{\text{degen}} H_{\text{ns}} P_{\text{degen}}$ have the form [39] (dropping the right-action indices)

$$|0, r, \{w\}\rangle = |0\rangle_{\text{sing}} \otimes \sum w_{\alpha^{(0)}} \left|r, \alpha^{(0)}\right\rangle\,. \tag{59}$$

The distribution over zero-weight states has to be found by solving the eigenvalue equation

$$H_{\text{ns}} |0, r, \{w\}\rangle = \Delta E\, |0, r, \{w\}\rangle\,. \tag{60}$$

We proceed to do (approximately) this.

## 4.2 A state with the KT energy

We construct the state of interest in three steps. First, in §4.2.1 we review the calculation of the energy in the adjoint sector [16,32,38,39]. We then argue that, at finite $N$, a straightforward extension of these methods to a state with a gas of $1 \ll \Bbbk \ll N$ weakly-interacting adjoints gives an energy much higher than the KT value (21). This failure to reproduce the KT energy arises from a trade-off between keeping the momenta of each adjoint low and minimising the interactions between them; we find that reducing one in a straightforward way increases the other. An important peculiarity of our approach is that we keep $N$ finite for much of the discussion, taking it to $\infty$ only at the end; the state of interest is more clearly isolated at finite $N$.

We then show that a 'liquid' state in which we 'clump' these $\Bbbk$ adjoints together has a much lower energy in §4.2.2; however, its energy is still higher than the KT value. We describe this as a liquid state because although the adjoints have condensed together, there is no appreciable barrier for one or more adjoints to 'spill' out and leave the clump.

To get the KT energy, it turns out that we need to 'solidify' this clump, i.e. make it so that there is a large barrier for an adjoint to leave. We solidify it by modifying the eigenvalue distribution in a radical way by creating a large $\mathcal{O}(\Bbbk^{1/4})$ hole in it; since the Fermi surface of the eigenvalue distribution is the spatial direction in the string theory, and since we are positing breaking it up into two disconnected pieces, we call this a "hole-in-the-world state." This is a solid because any adjoint that wants to leave has to tunnel across the hole in the world — this has a large cost and thus prevents adjoints from spilling out. This is the content of §4.2.3.

### 4.2.1 A gas of adjoints

**A single adjoint**
We begin by reviewing the calculation in the adjoint ($\Bbbk = 1$) sector [16, 32, 38, 39]. The

---

[12]The RHS should really be $N/2$, since the potential we have written down has two wells; this fact will not matter in our discussion.

zero-weight states are given by $\sum_k w_k \left| \boxed{k \; k} \right\rangle$, with $\sum_k w_k = 0$. The action of the algebra generators on this state are $(i < j)$

$$T_{ij}\left| \boxed{k \; k} \right\rangle = \left(\delta_{ik} - \delta_{jk}\right)\left| \boxed{i \; j} \right\rangle + (i \leftrightarrow j),$$

$$\tilde{T}_{ij}\left| \boxed{k \; k} \right\rangle = \iota\left(\delta_{ik} - \delta_{jk}\right)\left| \boxed{i \; j} \right\rangle - (i \leftrightarrow j),$$

$$T_{ij}^2\left| \boxed{k \; k} \right\rangle = \tilde{T}_{ij}^2\left| \boxed{k \; k} \right\rangle = 2\left(\delta_{ik} - \delta_{jk}\right)\left| \boxed{i \; i} \right\rangle + (i \leftrightarrow j). \tag{61}$$

Thus, the eigenvalue equation becomes

$$\sum_{i \neq j} \frac{w_i - w_j}{(\lambda_i - \lambda_j)^2} |0\rangle_{\text{sing}} \otimes \left| \boxed{i \; i} \right\rangle = \Delta E\, |0, \text{adj}, \{w\}\rangle. \tag{62}$$

This can be rewritten as the integral equation

$$\Delta E\, w(\lambda) = \mathscr{P} \int_{\sqrt{2\mu_F}}^{\lambda_c} \rho(\lambda') \frac{w(\lambda) - w(\lambda')}{(\lambda - \lambda')^2} \, d\lambda', \quad \text{s.t.} \quad \int_{\sqrt{2\mu_F}}^{\lambda_c} d\lambda\, \rho(\lambda) w(\lambda) = 0, \tag{63}$$

where the $\mathscr{P}$ means that the integral is evaluated using a principal value prescription (which follows from the $i \neq j$ in the sum). As pointed out in [16], it is useful to shift to the following variables,

$$h(\lambda) \equiv \rho(\lambda) w(\lambda), \quad \text{and} \quad \tau \equiv \cosh^{-1} \frac{\lambda}{\sqrt{2\mu_F}} \xrightarrow{\lambda \gg \sqrt{2\mu_F}} \log \frac{\lambda}{\sqrt{\mu_F}} + \dots \tag{64}$$

Matching the normalization condition (58) in these variables then gives, $\tau_c \sim \frac{1}{2} \log(N/\mu_F)$. The eigenvalue equation then becomes [16]

$$\Delta E\, h(\tau) = \frac{1}{2}\left(\log \frac{N}{\mu_F} - 2\tau \coth \tau\right) h(\tau) - \frac{1}{4\pi} \mathscr{P} \int_{-\frac{1}{2}\log(N/\mu_F)}^{\frac{1}{2}\log(N/\mu_F)} d\tau' \frac{h(\tau')}{\sinh^2 \frac{\tau - \tau'}{2}}. \tag{65}$$

This last term can be thought of as a kinetic term, whereas the rest can be thought of as a potential term. The kinetic term can be Fourier transformed to give $\frac{1}{\sqrt{2\pi^3}}(1 + \pi p \coth \pi p) h(p)$. This eigenvalue equation was explicitly solved in [48]; we will not use the details of this solution.

Taking $\lambda \gg \sqrt{\mu_F}$, all the coths can be approximated by 1 and this Hamiltonian becomes just

$$H_{1\text{pt}} = \frac{1}{2\pi} \log \frac{N}{\lambda^2} + \frac{1}{\sqrt{2\pi}} |\hat{p}|, \tag{66}$$

where $\hat{p}$ is the momentum operator that we defined implicitly by the Fourier transform of the last term in (65), and we have gone back from $\tau$ to $\lambda$.

**A gas of adjoints**

We first consider states where the distribution of matrix eigenvalues is perturbatively close to the singlet sector ground state, and where the adjoints are well-spread out and weakly interacting, as in figure 6. To be more explicit about what we mean by interaction, consider the effect of $T_{ij}^2 + \tilde{T}_{ij}^2$ on a state with multiple fundamentals and anti-fundamentals. Because the Hamiltonian comes with projectors onto the zero-weight subspace, we only consider the branches of the action that result in terms within the zero-weight subspace.

We take YTs of size 2 as an example. Suppose we take the YT

$$\boxed{\begin{array}{cc} i & i \; j \\ j & \end{array}} \tag{67}$$

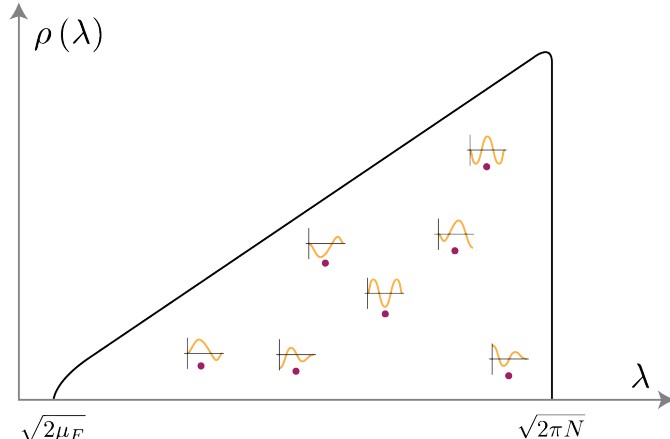

Figure 6: A gas of adjoints. We have drawn an adjoint at eigenvalue $\lambda$ as a purple dot in the position in the Fermi sea where the corresponding eigenvalue is. By a gas, we mean that the wavefunction approximately factorises into a product of wavefunctions of individual adjoints. The wavefunction above each dot is placed to remind the reader that the adjoints are not localised.

Defining

$$|T_{ij}|^2 = \frac{T_{ij}^2 + \tilde{T}_{ij}^2}{4}, \tag{68}$$

when $k \neq i, j$ we have

$$|T_{ik}|^2 \left| \begin{array}{ccc} i & i & j \\ j \end{array} \right\rangle = \left| \begin{array}{ccc} i & i & j \\ j \end{array} \right\rangle - \left| \begin{array}{ccc} k & k & j \\ j \end{array} \right\rangle ,$$

$$|T_{jk}|^2 \left| \begin{array}{ccc} i & i & j \\ j \end{array} \right\rangle = \left| \begin{array}{ccc} i & i & j \\ j \end{array} \right\rangle - \left| \begin{array}{ccc} i & i & k \\ k \end{array} \right\rangle . \tag{69}$$

We notice that the action of any generator that rotates an index appearing within the YT to one not appearing within the YT decouples into independent action on each box-antibox pair that share an index. We can therefore treat these terms as $\Bbbk$ copies of $H_{1\text{pt}}$ in (66).

We now analyze the action of $|T_{ij}|^2$ where both indices $i, j$ appear in the YT. For simplicity, we first consider a YT where the boxes labeled $i$ and $j$ share neither a row nor a column.

$$|T_{ij}|^2 \left| \begin{array}{cccc} i & k & k & i \\ j & j \end{array} \right\rangle = \left| \begin{array}{cccc} i & k & k & i \\ j & j \end{array} \right\rangle - \left| \begin{array}{cccc} j & k & k & j \\ j & j \end{array} \right\rangle$$

$$+ \left| \begin{array}{cccc} i & k & k & i \\ j & j \end{array} \right\rangle - \left| \begin{array}{cccc} i & k & k & i \\ i & i \end{array} \right\rangle \tag{70}$$

$$+ \left| \begin{array}{cccc} j & k & k & i \\ i & j \end{array} \right\rangle + \left| \begin{array}{cccc} i & k & k & j \\ j & i \end{array} \right\rangle .$$

Taking apart this equation, we see that the first two lines on the RHS correspond to the $H_{1\text{pt}}$ action we would expect from $|T_{ij}|^2$ on the $i$ boxes and the $j$ boxes respectively. The third line contains a new action — one that exchanges the $i$ and $j$ anti-boxes in the first term and the $i$ and $j$ boxes in the second term. Thus, the last line is generated by an exchange operator, which we denote $\jmath_{ij}$, and suggests that the interaction term between boxes is given by the

'cross' term

$$H_\times \equiv \sum_{i,j} \frac{\delta_{ij}}{(\lambda_i - \lambda_j)^2} \,. \tag{71}$$

We now check that this works for states where $i$ and $j$ share either a row or a column. Returning to two-box representations for simplicity, we find

$$|T_{ij}|^2 \left| \begin{array}{|c|c|} \hline i & i \\ \hline j & j \\ \hline \end{array} \right\rangle = 0 = \left| \begin{array}{|c|c|} \hline i & i \\ \hline j & j \\ \hline \end{array} \right\rangle - \left| \begin{array}{|c|c|} \hline j & j \\ \hline j & j \\ \hline \end{array} \right\rangle$$
$$+ \left| \begin{array}{|c|c|} \hline i & i \\ \hline j & j \\ \hline \end{array} \right\rangle - \left| \begin{array}{|c|c|} \hline i & i \\ \hline i & i \\ \hline \end{array} \right\rangle \tag{72}$$
$$+ \left| \begin{array}{|c|c|} \hline j & i \\ \hline i & j \\ \hline \end{array} \right\rangle + \left| \begin{array}{|c|c|} \hline i & j \\ \hline j & i \\ \hline \end{array} \right\rangle \,.$$

The LHS is equal to zero because the columns comprising the boxes and antiboxes are singlets under rotations from $i$ to $j$. The RHS is equal to zero due to the antisymmetry properties of the YT states. States corresponding to YTs with two $i$'s/$j$'s in the same column vanish, and the states appearing in the third row on the RHS cancel the first two states of the first two lines due to the anti-symmetrisation. We may again decompose this action into the sum of the $H_{1\text{pt}}$ action and an exchange operator, neither of which vanishes separately. A similar analysis can be done for the other possible combinations we can have of boxes sharing rows and columns, and we find that we may decompose the non-singlet Hamiltonian into

$$H_{\text{ns}} = H_{\text{dir}} + H_\times \,, \tag{73}$$

where $H_{\text{dir}}$ is the sum of $\Bbbk$ copies of $H_{1\text{pt}}$, and $H_\times$ is given by (71).

With this decomposition of the Hamiltonian in hand, we now search for wavefunctions of the labels of the dominant representation (figure 5) that match the KT prediction for the energy. We first present several general types of non-singlet excitation configurations on the singlet saddle point eigenvalue distribution and demonstrate why they fail. Then, we will resolve these confusions by showing that a certain class of states introduces a violent deformation of the eigenvalue configuration which precisely lowers the energy enough to match the KT result.

We first consider wavefunctions where the labels are taken to be well-separated, i.e., the maximum label that is allowed to appear in the YT is $i_{\text{max}} \gg \Bbbk$ — this allows us to neglect the cross-term to leading order. This state can be likened to a gas of adjoints. Due to the antisymmetrisation rules of the YT, we need at least $\mathcal{O}(\sqrt{\Bbbk})$ different labels for its construction. Since $\tau$ takes values in a range of width $\log N$, the spectrum of the momentum operator $\hat{p}$ has a spacing of $\mathcal{O}(1/\log N)$. Pauli exclusion then pushes labels up into states with a typical momentum of at least $p \sim \frac{\Bbbk^{1/2}}{\log N}$. The energy of such a configuration then scales as

$$E_{\Bbbk} = \frac{\Bbbk}{2\pi} \log N + \mathcal{O}\left( \frac{\Bbbk^{3/2}}{\log N} \right) \,. \tag{74}$$

The leading term in the above expression comes from the $\propto \log N$ term in $H_{\text{dir}}$ (66). The $\propto \log \lambda^2$ term can be ignored because of a subtlety in the order with which we take limits–for a fixed $N$, we populate the gas by choosing $\Bbbk$ ($\ll N^2$) eigenvalues to label the adjoints. We then take the large $N$ limit, whilst keeping these labels fixed. Thus the contribution from the $\propto \log \lambda^2$ term does not scale with $N$ and can be ignored to leading order.

If we assume $\Bbbk^{1/2} \leq \mathcal{O}(\log N)$ we can certainly find states with the KT energy where the non-singlet excitations are spread out as described above.[13] However, once $\Bbbk$ crosses this

---

[13]The minimum spreading required goes to the entire eigenvalue distribution as $\sqrt{\Bbbk} \to \log N$.

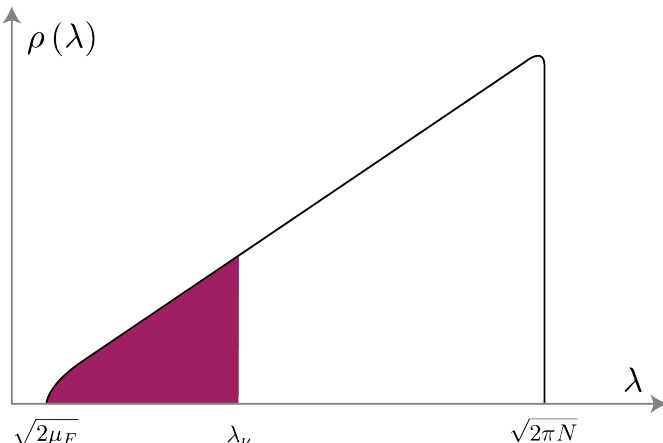

Figure 7: We can reduce the energy. at finite $N$, by clumping the adjoints together inside the Fermi sea as shown here.

bound, the kinetic energy on its own is much higher than the KT energy. Indeed, in the the double-scaling limit of interest, $\Bbbk = \mathcal{O}(N^0)$ and so such a state likely exists. However, there is nothing special about the state with energy $(2\pi)^{-1}\Bbbk \log N/\sqrt{\Bbbk}$ and we have no reason to believe it will dominate the partition function.

As a result, we are forced to look for other classes of states where the momentum and interaction cancel each other (the potential energy is strictly positive and can't cancel the kinetic energy). This is what we turn to now.

### 4.2.2 Condensing the adjoints into a liquid

We now show how to use the interactions to cancel the large momentum contributions and achieve a state with a much lower energy. Our strategy to do this is to kill significant subsets of the $U(N)$ generators by considering states that are singlets under subgroups of $U(N)$. In analogy with the 'gas' of adjoints that we described described earlier, we think of this as a 'liquid' solution, where the adjoints are clumped together as in Figure 7 but there is no significant barrier against them 'spilling out' of the clump.[14] This is an intermediate step in our analysis – as we will see, this reduces the energy but still gives a larger value than the KT energy (21); this 'liquid' state never dominates the partition function and should not be regarded as an intermediate phase.

The interaction term between a box-antibox pair with index $i$ and one with index $j$ is the positive operator $|T_{ij}|^2/(\lambda_i - \lambda_j)$. Thus the way we minimise its contributions is to construct a state that is annihilated by as many of the operators $T$, $\tilde{T}$ as possible. As a start, we first consider YTs with all indices taken from the set $\{1, \ldots, \nu\}$ — this ensures that all the generators with *both* indices larger than $\nu$ kill the state. Further, as we explain below, we can ensure this state is an $U(\nu)$ singlet — this results in all the generators with *both* indices in $\{1, \ldots, \nu\}$ also annihilating it. Thus, the state that we are after is a singlet of the subgroup $U(\nu) \times U(N - \nu)$. We note that if we are interested in the approximately isosceles YD that dominates the entropy, we need

$$\nu \geq \sqrt{2\Bbbk}, \tag{75}$$

since this is the size of the largest (anti-)column (see figure 5) and different boxes in a column must have distinct indices (otherwise they can't be anti-symmetrised).

---

[14]Only a few adjoints can spill out without a high energy cost.

Now we turn to constructing a $U(\nu)$ singlet. Such a singlet in the adjoint sector is simply a trace over the subset of indices,

$$|\nu\rangle \equiv \frac{1}{\sqrt{\nu}} \sum_{i=1}^{\nu} \left| \boxed{i}\,\boxed{i} \right\rangle \quad \leftrightarrow \quad \frac{1}{\sqrt{\nu}} \begin{pmatrix} \mathbb{1}_{\nu\times\nu} & 0_{N-\nu\times\nu} \\ 0_{\nu\times N-\nu} & 0_{N-\nu\times N-\nu} \end{pmatrix}, \tag{76}$$

where the matrix on the right is the wavefunction written as a two-index tensor.[15] The state we are interested in is $\Bbbk$ copies of this $U(\nu)$ singlet (76), projected into the irrep of interest.

That the state remains a $U(\nu)$ singlet after the projection is clear by the fact that the generators act irreducibly within each irrep, and so the state can only be annihilated by any generator if each projection is; schematically, for any generator $T$ we have

$$\left(e^{\iota T_{\text{adj}}}\right)^{\otimes\Bbbk} |\psi\rangle = \bigoplus_{r\in\text{adj}^{\otimes\Bbbk}} e^{\iota T_r} |\psi\rangle = 1 \quad \Rightarrow \quad T_r P_r |\psi\rangle = 0. \tag{77}$$

Further, the projection is unnecessary, since we know from the discussion in §3.2 that a single irrep dominates the tensor product of $\Bbbk$ adjoints. One can check using the same techniques that the dimension of the zero-weight subspace of the dominant irrep is $\binom{N}{\Bbbk} \cdot \left(\sqrt{\Bbbk!}\right)^2 = N^{\Bbbk}$, which is the dimension of $\Bbbk$ zero-weight adjoints. Thus, the the $U(\nu)\times U(N-\nu)$ singlet of interest is merely,

$$|\nu,\Bbbk\rangle \equiv |\nu\rangle^{\otimes\Bbbk} = \left(\frac{1}{\sqrt{\nu}} \sum_{i=1}^{\nu} \left| \boxed{i}\,\boxed{i} \right\rangle\right)^{\otimes\Bbbk}, \tag{78}$$

and the liquid state is therefore given by,

$$|0_l\rangle \equiv |\rho(\lambda)\rangle \otimes |\nu,\Bbbk\rangle. \tag{79}$$

Written in terms of YTs in the dominant irrep, this state is a uniform superposition over all reflection-symmetric (i.e. self-conjugate) YTs with indices in $\{1,\ldots,\nu\}$.[16]

We can also check this last point explicitly by calculating the value of the second Casimir and its standard deviation in the state. The second Casimir is

$$C_2 = \sum_{i<j} \frac{T_{ij}^2 + \tilde{T}_{ij}^2}{2} + \sum_i H_i^2. \tag{80}$$

The only generators that don't annihilate the state are those with $i < \nu, j > \nu$; and each of these can act on any of the $2\Bbbk$ (anti-)boxes. Its expectation value in the state above is

$$\langle 0_l| C_2 |0_l\rangle = \frac{2\Bbbk}{\nu} \sum_{i\leq\nu, j>\nu} 1 \approx 2\Bbbk N + \mathcal{O}(\Bbbk\,\nu), \tag{81}$$

which can easily be checked to be the value of the Casimir in the dominant irrep of Figure 5, using standard formulae found in e.g. [49]. However, this is not enough; we must also check the variance to make sure that it has a small spread in irreps. By a similar logic as above, we find

$$\langle C_2^2\rangle_l \approx \langle C_2\rangle_l^2, \tag{82}$$

---

[15]This state isn't exactly in the addjoint irrep – for instance, it has a non-zero inner product with the state $\mathbb{1}_{N\times N}/\sqrt{N}$, which is a $U(N)$ singlet. However this inner product is $\sqrt{\nu/N} \ll 1$, and so we can safely ignore this subtlety.

[16]A zero-weight state in any fixed irrep can also be written as a tensor $T^{i_1\ldots i_{\Bbbk}}_{i_{\sigma(1)}\ldots i_{\sigma(\Bbbk)}}$ where $\sigma$ is a permutation in $S_{\Bbbk}$ and the symmetry properties of the upper (lower) indices are dictated by the shape of the YD (anti-YD). This tensor is the wavefunction on the space of YTs. The wavefunction of $|\nu,\Bbbk\rangle$ is zero if any of the indices $i_1\cdots i_{\Bbbk}$ is bigger than $\nu$ or if the permutation $\sigma$ is not the identity element of $S_{\Bbbk}$, and 1 (up to a normalisation) otherwise.

and the variance is therefore subleading. All of this is concomitant with our result in §3 that one irrep dominates the product of many adjoints.

The expectation value of the non-singlet Hamiltonian in this state is

$$\langle H_{\text{ns}}\rangle_l \equiv \langle 0_l|H_{\text{ns}}|0_l\rangle = \frac{1}{\nu^{\Bbbk}}\left(\sum_{i=1}^{\nu}\Big\langle\,\boxed{\,i\;\;i\,}\,\Big|\right)^{\otimes\Bbbk}\left(\frac{1}{4}\sum_{j<k}\frac{T_{jk}^2+\tilde T_{jk}^2}{(\lambda_j-\lambda_k)^2}\right)\left(\sum_{i'=1}^{\nu}\Big|\,\boxed{\,i'\;\;i'\,}\,\Big\rangle\right)^{\otimes\Bbbk}. \tag{83}$$

We need only consider the case when $j<\nu$, $k>\nu$, because the state is annihilated by all the other generators. The action that contributes to this expectation value is the one where both operators in each $T^2$ act on the same (anti-)box. This gives

$$\langle H_{\text{ns}}\rangle_l = \frac{\Bbbk}{\nu}\sum_{j=1}^{\nu}\sum_{k=\nu+1}^{N}\frac{1}{(\lambda_j-\lambda_k)^2} = \frac{\Bbbk}{\nu}\int_0^{\lambda_\nu}\mathrm{d}\lambda'\int_{\lambda_{\nu+1}}^{\lambda_N}\mathrm{d}\lambda\,\frac{\rho(\lambda')\rho(\lambda)}{(\lambda-\lambda')^2}. \tag{84}$$

Here, $\lambda_i$ is the value of the $i^{th}$ eigenvalue, which is the solution of the equation

$$\int_0^{\lambda_i}\rho(\lambda)\,\mathrm{d}\lambda = i. \tag{85}$$

Let us plug in now the vacuum eigenvalue density $\rho(\lambda)=\lambda/\pi$. We find

$$\langle H_{\text{ns}}\rangle_l = \frac{\Bbbk}{2\pi^2\nu}\Big[\big[\lambda\,\lambda'+\big(\lambda^2+\lambda'^2\big)\log|\lambda-\lambda'|\big]_0^{\lambda_\nu}\Big]_{\lambda_{\nu+1}}^{\lambda_c}$$

$$\approx \frac{\Bbbk}{2\pi^2\nu}\Big\{-\lambda_\nu\lambda_{\nu+1}+\lambda_\nu^2\log\lambda_c-\big(\lambda_\nu^2+\lambda_{\nu+1}^2\big)\log(\lambda_{\nu+1}-\lambda_\nu)+\lambda_{\nu+1}^2\log\lambda_{\nu+1}\Big\}. \tag{86}$$

To simplify this further, we have to use the facts that $\lambda_i=\sqrt{2\pi i}$ and $\lambda_{\nu+1}-\lambda_\nu\sim\sqrt{\frac{\pi}{\nu}}$ in the vacuum eigenvalue density. Thus we find

$$\langle H_{\text{ns}}\rangle_l = \frac{\Bbbk}{2\pi}\left(\log N+\frac{3}{2}\log\Bbbk\right)+\mathcal{O}(\Bbbk). \tag{87}$$

The UV divergence coming from $\lambda_{\nu+1}$ approaching $\lambda_\nu$, while of the same magnitude as the subleading term in the KT energy, causes the subleading term to have the wrong sign. The UV divergent term is the penultimate one $-\big(\lambda_\nu^2+\lambda_{\nu+1}^2\big)\log(\lambda_{\nu+1}-\lambda_\nu)$; this is a positive quantity because the argument of logarithm is small. This divergence is the last remnant of the large kinetic energy discussed in the previous section, and it seems we cannot push this contribution lower without drastically modifying the structure of the UV cutoff in the emergent geometry.

In summary, in our efforts to use the interaction between the non-singlet excitations to eliminate as many of the $|T_{ij}|^2$'s in the Hamiltonian as possible, we were forced to introduce a sharp cutoff in the non-singlet distribution. This sharp cutoff functions as a domain wall between a region dense with boxes and a region with no boxes, and the UV divergence can then be thought of as a domain wall energy.

We have not been able to conclusively prove that there is no state in the unperturbed eigenvalue distribution that matches the KT energy in the regime where $\Bbbk\gg\log N$. There may still be room for careful interplay between the cross term and direct term in the non-singlet Hamiltonian to cancel the large kinetic energies seemingly forced on us by Pauli exclusion. However, the methods we have considered have not yielded a way past this obstruction. In the next section we show how allowing the domain wall described above to violently backreact on the background eigenvalue distribution, ripping a hole in the emergent geometry, eliminates the UV divergences and lowers the energy to precisely match the KT result.

### 4.2.3 Freezing the liquid by blowing up a hole in the world

We now show that there is always a bound state for $1 \ll \Bbbk \ll N^2$ — the same regime where the entropy calculation is valid — whose energy matches the KT value (21). This state is one where the non-singlet excitations have backreacted strongly on the eigenvalues, creating a large gap in the eigenvalue distribution. Since the bulk spacetime on which the string theory lives is formed by the eigenvalue Fermi sea (up to a non-local integral transform), this configuration is like a hole in the world; we call it the hole-in-the-world state.[17] Blowing up a hole in the world naïvely seems to be in contradiction with the string theory, but we argue that this hole is exactly what is needed for a duality with sine-Liouville string theory in §6.1.1.

The hole-in-the-world solution can be motivated by the following numerological observation. Rearrange the terms of (86) as

$$E_{\rm ns} = \frac{\Bbbk}{2\pi^2 \nu} \left\{ -\lambda_\nu \lambda_{\nu+1} + \lambda_\nu^2 \log \lambda_c - \lambda_\nu^2 \log(\lambda_{\nu+1} - \lambda_\nu) - \lambda_{\nu+1}^2 \log\left(1 - \frac{\lambda_\nu}{\lambda_{\nu+1}}\right) \right\}. \tag{88}$$

The last term here gave a large positive contribution to (87) because $\lambda_{\nu+1} - \lambda_\nu \ll 1$ in a smooth eigenvalue distribution, and this made the energy exceed the KT value (21). Numerologically, these terms can add up to the KT energy if we allow the eigenvalues $\lambda_{\nu+1}$ and $\lambda_\nu$ to differ by a large amount,

$$\lambda_{\nu+1} - \lambda_\nu = \lambda_\nu \cdot \mathcal{O}(1) = \Bbbk^{1/4} \cdot \mathcal{O}(1) \implies \langle H_{\rm ns} \rangle_l \longmapsto \frac{\Bbbk}{2\pi} \log \frac{N}{\sqrt{\Bbbk}} + \mathcal{O}(\Bbbk). \tag{89}$$

Here we've again taken $\nu = \mathcal{O}(\sqrt{\Bbbk})$ and $\lambda_\nu, \lambda_{\nu+1} = \mathcal{O}(\Bbbk^{1/4})$. This is a radical change in the eigenvalue distribution, demanding that there are no eigenvalues for an $\mathcal{O}(\Bbbk^{1/4})$ region of eigenvalue space, as shown in figure 8. Denoting the new eigenvalue distribution by $\rho_n(\lambda)$, the hole-in-the-world state is given by

$$|0\rangle \equiv |\rho_n(\lambda)\rangle \otimes |\nu, \Bbbk\rangle. \tag{90}$$

The remainder of this section is devoted to arguing that there is indeed an approximate eigenstate of the form (90) that satisfies (89). Since this state differs radically from other considered previously in the literature, we should treat it with care. We begin with a more general variational ansatz and show that the variational principle leads to the hole-in-the-world state we described above. This is a rather technical exercise and readers interested only in the main physics points may safely skip to §4.3.

We now modify the eigenvalue density to

$$\rho_n(\lambda) = \frac{1}{\pi} \begin{cases} \lambda + n'(\lambda), & 0 < \lambda < \lambda_1, \\ 0, & \lambda_1 < \lambda < \lambda_2, \\ \lambda + n(\lambda), & \lambda > \lambda_2. \end{cases} \tag{91}$$

The functions $n, n'$ are constrained by

$$\int_0^{\lambda_c} \rho_n(\lambda) \, d\lambda = N \implies \frac{\lambda_2^2 - \lambda_1^2}{2} = \int_0^{\lambda_1} n'(\lambda) \, d\lambda + \int_{\lambda_2}^{\lambda_c} n(\lambda) \, d\lambda. \tag{92}$$

We will minimise the energy over the parameters $\lambda_1, \lambda_2$ and the functions $n, n'$. We assume that the maximum value of the eigenvalue $\lambda_c = \sqrt{2\pi N}$ is not modified.

---

[17]This might not be precisely a hole in the string theory, due to the non-local integral transformation relating the eigenvalue and dilaton directions. However, as we discuss in §6.1.1, there are indications of hole-like behaviour in the string side as well. We argue that the scattering and bound states of sine-Liouville theory are excitations to the right and left of the hole respectively.

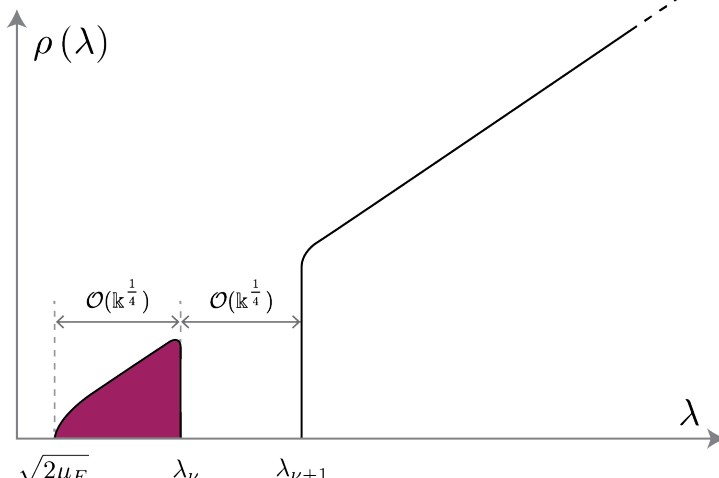

**Figure 8:** A visual representation of the hole-in-the-world state, showing the new eigenvalue density (91). The coloured eigenvalue are host to non-singlet excitations and the uncoloured ones are unoccupied. $\lambda_\nu$, $\lambda_{\nu+1}$, and $\lambda_{\nu+1} - \lambda_\nu$ are all $\mathcal{O}(\Bbbk^{1/4})$. Here, we have taken the double-scaling limit $N \to \infty$.

To calculate the energy of these contributions, we first review the calculation of the energy in the singlet vacuum, using the phase space density as in (56). The eigenvalue density is given by integrating over the momentum, and the energy by integrating the Hamiltonian against this phase space density. Setting $\mu_F = 0$, which we are now allowed to do, the integral is over $|p| \le \lambda$.

The energy difference between the new and the old eigenvalue distributions is

$$\Delta E_{\mathrm{eig}} = \frac{1}{2\pi} \int_0^{\lambda_c} \mathrm{d}\lambda \cdot 2 \int_{\pi\rho(\lambda)}^{\pi\rho_n(\lambda)} \mathrm{d}p \, \frac{p^2 - \lambda^2}{2} \, . \tag{93}$$

The factor of 2 comes from the fact that we have to sum over both positive as well as negative momenta. We find

$$\Delta E_{\mathrm{eig}} = \frac{1}{2\pi} \int_0^{\lambda_1} \mathrm{d}\lambda \, n'^2 \left( \frac{n'}{3} + \lambda \right) + \frac{\lambda_2^4 - \lambda_1^4}{12\pi} + \frac{1}{2\pi} \int_{\lambda_2}^{\lambda_c} \mathrm{d}\lambda \, n^2 \left( \frac{n}{3} + \lambda \right) . \tag{94}$$

The term in the middle is the cost of scooping out the eigenvalues to create the hole — it is a positive term because the original energy of these eigenvalues was negative.

The non-singlet energy with this new density of states is found by plugging in the new density of states into (84). This gives

$$E_{\mathrm{ns}} \equiv \langle 0 | H_{\mathrm{ns}} | 0 \rangle = \frac{\Bbbk}{\pi^2 \nu} \int_0^{\lambda_1} \mathrm{d}\lambda' \int_{\lambda_2}^{\lambda_c} \mathrm{d}\lambda \, \frac{(\lambda' + n')(\lambda + n)}{(\lambda - \lambda')^2} , \qquad \nu = \int_0^{\lambda_1} \rho_n(\lambda) \, \mathrm{d}\lambda . \tag{95}$$

The $n$, $n'$-independent part was discussed in (89), modulo the fact that the scalings taken there have not yet been justified. The $n$-dependent part of this is

$$E_{\mathrm{ns}}\big|_n = \frac{\Bbbk}{\pi^2 \nu} \int_{\lambda_2}^{\lambda_c} \mathrm{d}\lambda \, n(\lambda) \left[ \log\left( 1 - \frac{\lambda_1}{\lambda} \right) + \frac{\lambda_1}{\lambda - \lambda_1} \right] + \frac{\Bbbk}{\pi^2 \nu} \int_0^{\lambda_1} \mathrm{d}\lambda' \int_{\lambda_2}^{\lambda_c} \mathrm{d}\lambda \, \frac{n(\lambda) n'(\lambda')}{(\lambda - \lambda')^2} , \tag{96}$$

and its $n'$-dependent part is

$$E_{\mathrm{ns}}\big|_{n'} = \frac{\Bbbk}{\pi^2 \nu} \int_0^{\lambda_1} \mathrm{d}\lambda' \, n'(\lambda') \left[ \log \frac{\lambda_c}{\lambda_2 - \lambda'} + \frac{\lambda'}{\lambda_2 - \lambda'} \right] + \frac{\Bbbk}{\pi^2 \nu} \int_0^{\lambda_1} \mathrm{d}\lambda' \int_{\lambda_2}^{\lambda_c} \mathrm{d}\lambda \, \frac{n(\lambda) n'(\lambda')}{(\lambda - \lambda')^2} . \tag{97}$$

Because of the large $\log \lambda_c \sim \log N$ term in (97), minimisation requires that we set

$$\int_0^{\lambda_1} n'(\lambda')\, d\lambda' = 0\,. \tag{98}$$

In other words, all the eigenvalues we scooped out between $\lambda_1$ and $\lambda_2$ get sent to larger eigenvalues — this makes sense since the repulsion wants to send them far away and there's a lot more space at larger eigenvalues. The ones to the left of the hole can move around but stay to the left of the hole. In particular, this means that

$$\lambda_1 = \lambda_\nu = \sqrt{2\pi\nu}\,, \tag{99}$$

as was the case even in the old eigenvalue distribution.

To minimise the energy, we have to minimise the 'action'

$$E = \Delta E_{\text{eig}} + E_{\text{ns}} - \frac{\alpha}{2\pi}\left(\int_{\lambda_2}^{\lambda_c} n(\lambda)\, d\lambda - \frac{\lambda_2^2 - \lambda_1^2}{2}\right) - \frac{\alpha'}{2\pi}\int_0^{\lambda_\nu} n'(\lambda')\, d\lambda'\,, \tag{100}$$

where $\alpha, \alpha'$ are Lagrange multipliers. Setting $\delta E/\delta n, \delta E/\delta n'$ to 0 gives

$$n(\lambda) = \sqrt{\lambda^2 + \alpha - \beta(\lambda)} - \lambda\,,$$
$$n'(\lambda') = \sqrt{\lambda'^2 + \alpha' - \beta'(\lambda')} - \lambda'\,, \tag{101}$$

where

$$\frac{\beta(\lambda)}{2\pi} = \frac{\Bbbk}{\pi^2\nu}\left\{\log\left(1 - \frac{\lambda_1}{\lambda}\right) + \frac{\lambda_1}{\lambda - \lambda_1} + \int_0^{\lambda_1} d\lambda'\, \frac{n'(\lambda')}{(\lambda - \lambda')^2}\right\}\,,$$
$$\frac{\beta'(\lambda')}{2\pi} = \frac{\Bbbk}{\pi^2\nu}\left\{-\log\left(\lambda_2 - \lambda'\right) + \frac{\lambda'}{\lambda_2 - \lambda'} + \int_{\lambda_2}^{\lambda_c} d\lambda\, \frac{n(\lambda)}{(\lambda - \lambda')^2}\right\}\,. \tag{102}$$

Now we have to plug in these solutions into the constraints to solve for the Lagrange multipliers. The constraint for $n$ is easier to deal with, because the integral has a large IR region $\lambda \gg \lambda_{1,2}$. At large $\lambda$, the $n'$-dependent term in $\beta$ is effectively proportional to $\int d\lambda'\, n'(\lambda') = 0$, up to corrections that scale with $\lambda_1^2/\lambda^2$, and the $1/\lambda$ contributions from the first two terms cancel each other, giving $\beta(\lambda) \sim \Bbbk/\nu \cdot \lambda_1^2/\lambda^2$. Thus, at large $\lambda$, we find that $n \sim \alpha/2\lambda$. Plugging this back into the constraint to solve for $\alpha$ gives[18]

$$\alpha = \frac{\lambda_2^2 - \lambda_1^2}{\log \lambda_c} \quad\Longrightarrow\quad n \approx \lambda\left\{\sqrt{1 - \frac{\beta(\lambda)}{\lambda^2}} - 1\right\}\,. \tag{103}$$

Close to the hole, this quantity is negative and we have thus found that the non-singlet interaction pushes the eigenvalues away from the hole. Far away from the hole, it vanishes. Since $\lambda_1/\lambda_2 < 1$, we expand in $\lambda_1/\lambda$ to find

$$\frac{\beta(\lambda)}{2\pi} \approx \frac{\Bbbk}{2\pi^2\nu}\frac{\lambda_1^2}{\lambda^2}\,, \qquad n \approx -\frac{\Bbbk}{4\pi^2\nu}\frac{\lambda_1^2}{\lambda^3} = -\frac{\Bbbk}{2\pi\lambda^3}\,. \tag{104}$$

Plugging this into (102) and expanding in $\lambda'/\lambda_2$ gives

$$\frac{\beta'}{2\pi} \approx -\frac{\Bbbk}{\pi^2\nu}\log\lambda_2\,, \qquad n' \approx \sqrt{\lambda'^2 + \alpha' + \frac{\Bbbk\log\lambda_2}{\pi^2\nu}} - \lambda'\,. \tag{105}$$

---

[18]Here, we are assuming that $\log N \gg \sqrt{\Bbbk}$, consistent with our interest in the double-scaling limit. It can be checked, by not dropping the $\alpha$ term in this equation, that this is not necessary for the main result of this section to hold. We will use this fact in the discussion to follow.

The constraint (98) can only be satisfied if

$$n' = 0.\tag{106}$$

It is straightforward to plug (103) and (106) back into the energy (100) to find

$$E = \frac{\lambda_2^4 - \lambda_1^4}{12\pi} + a\,\frac{\Bbbk^2}{\nu^2}\frac{\lambda_1^4}{\lambda_2^4} + \frac{\Bbbk}{2\pi^2\nu}\left[-\lambda_1\lambda_2 + \lambda_1^2\log\frac{\lambda_c}{\lambda_2 - \lambda_1} - \lambda_2^2\log\left(1 - \frac{\lambda_1}{\lambda_2}\right)\right].\tag{107}$$

Here, $a$ is an $\mathcal{O}(1)$ factor. To minimise with respect to $\lambda_2$, we set the derivative with respect to $\lambda_2$ to 0, giving the equation (setting $\lambda_1^2 = 2\pi\nu$ and not keeping track of coefficients)[19]

$$0 = a_1\frac{\Bbbk^2}{\lambda_2^5} + a_2\lambda_2^3 + a_3\frac{\Bbbk}{\sqrt{\nu}} + a_4\frac{\Bbbk}{\lambda_2 - \lambda_1} = \mathcal{O}\left(\Bbbk^{1/4}\right) \qquad \implies \qquad \sqrt{\nu}, \lambda_2 = \mathcal{O}\left(\Bbbk^{1/4}\right).\tag{108}$$

With this scaling the last three terms are all $\mathcal{O}(\Bbbk^{3/4})$ and the first term is smaller. We can now check with this scaling that the energy is

$$E = \frac{\Bbbk}{2\pi}\log\frac{N}{\sqrt{\Bbbk}} + \mathcal{O}(\Bbbk).\tag{109}$$

**Fluctuations in the energy**

Since this is not an exact eigenstate, we should also check that it is an approximate eigenstate of the Hamiltonian by calculating the standard deviation of the energy. Since the eigenvalue sector is in a classical limit (which is why we could use the language of eigenvalue densities), it doesn't contribute to the variance at leading order.

To calculate the variance due to the non-singlets, we first consider the action of the non-singlet Hamiltonian on one copy of the $U(\nu)$ singlet, $|\nu\rangle$,

$$H_{\text{ns}}|\nu\rangle \equiv |\nu'\rangle = \frac{1}{\sqrt{\nu}}\sum_{j\leq\nu, k>\nu}\frac{\left|\boxed{\phantom{x}j\phantom{x}}\right\rangle - \left|\boxed{\phantom{x}k\phantom{x}}\right\rangle}{(\lambda_j - \lambda_k)^2}$$

$$= \frac{1}{\sqrt{\nu}}\left\{\sum_{j\leq\nu}\underbrace{\sum_{k>\nu}\frac{1}{(\lambda_j - \lambda_k)^2}}_{\equiv\alpha_j}\left|\boxed{\phantom{x}j\phantom{x}}\right\rangle - \sum_{k>\nu}\underbrace{\sum_{j\leq\nu}\frac{1}{(\lambda_j - \lambda_k^2)^2}}_{\equiv\beta_k}\left|\boxed{\phantom{x}k\phantom{x}}\right\rangle\right\}.\tag{110}$$

Converting the $\alpha_j$ and $\beta_k$ to integrals give,

$$\alpha_j = \int_{\lambda_{\nu+1}}^{\lambda_c} d\lambda\,\frac{\rho_n(\lambda)}{(\lambda - \lambda_j)^2} = \frac{1}{\pi}\left\{\log\frac{\lambda_c}{\lambda_{\nu+1} - \lambda_j} + \frac{\lambda_j}{\lambda_{\nu+1} - \lambda_j}\right\},$$

$$\beta_k = \int_0^{\lambda_\nu} d\lambda\,\frac{\rho_n(\lambda)}{(\lambda - \lambda_k)^2} = \frac{1}{\pi}\left\{\log\left(1 - \frac{\lambda_\nu}{\lambda_k}\right) + \frac{1}{1 - \frac{\lambda_\nu}{\lambda_k}} - 1\right\}.\tag{111}$$

Acting on the hole-in-the-world state $|0\rangle$, the Hamiltonian gives

$$H_{\text{ns}}|0\rangle = |\rho_n(\lambda)\rangle \otimes \left(\sum_{r=1}^{\Bbbk}|\nu\rangle^{\otimes(r-1)} \otimes |\nu'\rangle \otimes |\nu\rangle^{\otimes(\Bbbk-r)}\right).\tag{112}$$

---

[19]We have dropped terms arising from the derivatives of the logarithms for brevity. Because we are only interested in the scaling of $\lambda_2$, it is easily checked that the addition of these terms don't change the result.

The expectation value of the Hamiltonian is

$$\langle H_{\mathrm{ns}}\rangle = \frac{\Bbbk}{\nu}\sum_j \alpha_j \equiv \Bbbk\,\bar{\alpha}\,. \tag{113}$$

Now we can calculate the two-point function of $H_{\mathrm{ns}}$

$$\langle 0|H_{\mathrm{ns}}^2|0\rangle = \|H_{\mathrm{ns}}\,|0\rangle\|^2 = \frac{\Bbbk}{\nu}\left(\sum_j \alpha_j^2 + \sum_k \beta_k^2\right) + 2\binom{\Bbbk}{2}\bar{\alpha}^2\,. \tag{114}$$

The first term proportional to $\Bbbk/\nu$ is from the terms where both copies of $H_{\mathrm{ns}}$ act on the same adjoint, and the term proportional to $\binom{k}{2}$ comes from the case when the two $H_{\mathrm{ns}}$'s act on different adjoints. The variance is

$$\langle H_{\mathrm{ns}}^2\rangle - \langle H_{\mathrm{ns}}\rangle^2 = \Bbbk\,\overline{(\alpha - \bar{\alpha})^2} + \frac{\Bbbk}{\nu}\sum_{k=\nu+1}^{N}\beta_k^2\,, \tag{115}$$

where the overline means the average over the first $\nu$ eigenvalues. The two terms can be calculated, as always, by converting the sum into an integral

$$\Bbbk\,\overline{(\alpha - \bar{\alpha})^2} \sim \Bbbk\,\frac{\lambda_\nu^2}{\nu} = \mathcal{O}(\Bbbk)\,, \tag{116}$$

$$\frac{\Bbbk}{\nu}\sum_{k=\nu+1}^{N}\beta_k^2 = \mathcal{O}(\Bbbk)\,, \tag{117}$$

$$\implies \qquad \langle H_{\mathrm{ns}}^2\rangle - \langle H_{\mathrm{ns}}\rangle^2 = \mathcal{O}(\Bbbk)\,. \tag{118}$$

Thus, the standard deviation is $\mathcal{O}\left(\sqrt{\Bbbk}\right)$, which is significantly smaller than both the energy and the corrections to it that we have not kept track of. We can then safely conclude that the hole-in-the-world state is an approximate eigenstate whose energy matches the energy found by KT (21).

## 4.3 The fixed-charge partition function

We have isolated a particular bound-state within the dominant $PU(N)$ irrep, so we have picked one state out of a possible $N^{\Bbbk}$.[20] Further the considerations in §4.2.1 and §4.2.2 together provide some (inconclusive) evidence that this is more or less the lowest energy state in the sector. However, our Hamiltonian derivation of the KT partition function is not yet complete. This is because this state may not dominate the partition function, since in general the interplay of energy and entropy can cause a higher energy state to dominate the partition function.

The hole-in-the-world state has two free parameters that the analysis in §4.2.3 did not fix, the position of the hole's edges $\lambda_1, \lambda_2$ and the modification of the eigenvalue distribution to its right $n(\lambda)$. All we found was that the former is $\mathcal{O}\left(\Bbbk^{1/4}\right)$ and the latter has the asymptotic form (103). Thus, the number of orthogonal hole-in-the-world states grows polynomially with $\Bbbk$, giving an entropy $\sim \log\Bbbk$ (apart from the large entropy discussed in §3). Thus, we have to check if there are any other states such that

$$S - 2\pi RE \;>\; -R\Bbbk\log\frac{N}{\sqrt{\Bbbk}} + \mathcal{O}(\Bbbk)\,, \qquad 1 < R < 2\,. \tag{119}$$

---

[20]We have factored out the degeneracy discussed in §3. The number $N^{\Bbbk}$ can be derived by the argument above (78).

The condition on $R$ comes from the string theory [11, 12]. Since $R$ is an $\mathcal{O}(1)$ number, all we need to check is that the energy cost of excitations outweighs the entropy cost. There are two sorts of states to check, excitations of the solid itself and states where we let all or part of the solid evaporate into the large eigenvalue region.

There are roughly $\sqrt{\Bbbk}^{\Bbbk}$ states that are excitations of the solid. The count of states is the dimension of $\Bbbk$ zero-weight $U(\nu)$ adjoints. Thus the entropic advantage here is $\sim \Bbbk \log \Bbbk$. To calculate the energy cost, we use the result (106) that the eigenvalue distribution doesn't get modified within the solid. We start with the observation that compressing $\Bbbk$ vortices into a region of size $\frac{1}{4} \log \Bbbk$ in $\tau$-space will lead to kinetic energies of order $\frac{\Bbbk^{3/2}}{\log \Bbbk}$. We now evaluate the variance due to the interactions around this large kinetic energy by noting that the exchange operators appearing in the numerators in the cross-term have precisely one positive and one negative eigenvalue of unit magnitude. We therefore model a generic contribution from the cross term as a random walk — each pair of boxes located at eigenvalues $i, j$ will randomly contribute either $(\lambda_i - \lambda_j)^{-2}$ or $-(\lambda_i - \lambda_j)^{-2}$ to the energy. The number of steps in the random walk will be the number of pairs of boxes — $\Bbbk^2/2$. The mean square of the step size is evaluated as

$$\frac{1}{\Bbbk} \int_0^{\Bbbk^{1/4}} \lambda \, d\lambda \int_0^{\Bbbk^{1/4}} \lambda' \, d\lambda' \frac{1}{(\lambda - \lambda')^4} \approx \mathcal{O}(\Bbbk^{1/4}). \tag{120}$$

Here the $1/\Bbbk$ factor comes from averaging over $\Bbbk$ pairs of eigenvalues inside the solid. The random walk approximation therefore implies that the standard deviation in the cross-term energy goes like

$$\Delta E_{\times} \sim \Bbbk \cdot \Bbbk^{1/8} \sim \mathcal{O}(\Bbbk^{9/8}). \tag{121}$$

Thus, we have determined that the typical energy in this class of states is $\mathcal{O}\left(\frac{\Bbbk^{3/2}}{\log \Bbbk}\right) \pm \mathcal{O}(\Bbbk^{9/8})$. This dominates the entropic contribution to the free energy of $\mathcal{O}(\Bbbk \log \Bbbk)$, showing the ground state we have picked out indeed dominates the partition function.

We can perform a similar analysis if we allow the box labels to range over the entirety of the eigenvalue range, from $\lambda = 0$ to $\lambda = \sqrt{2\pi N}$. There will be $\left(\frac{N}{\sqrt{\Bbbk}}\right)^{\Bbbk}$ such states, with a typical energy of order $\frac{\Bbbk^{3/2}}{\log N}$. The energy will again suppress the contribution from these states provided $\Bbbk \log N \ll \frac{\Bbbk^{3/2}}{\log N}$, i.e. when

$$\Bbbk \gg \log N. \tag{122}$$

We saw a similar but stronger condition in §4.2.1 as the condition that there are no gas of adjoints states with the KT value of the energy. Thus, we have found that the bound state above genuinely dominates the partition function at $\Bbbk \gg \log N$.

Unfortunately, however, (122) is not satisfied in the double-scaling limit, where we expect $\Bbbk \sim \mathfrak{z}^{\frac{4}{2-R}} = \mathcal{O}(N^0)$. This is natural, since in the double-scaling limit the black hole is not in a finite-volume box and so should evaporate, whereas when (122) *is* satisfied there is a box with finite size relative to the black hole. However, as we argue in the following section, the hole-in-the-world state is nevertheless metastable when $\Bbbk = \mathcal{O}(N^0)$, with a lifetime that grows polynomially in $\Bbbk$.

This leaves the question of why this state should dominate the partition function in the double-scaling limit. We have not been able to reach a definitive understanding of this question. If we calculate the partition function *after* taking the double-scaling limit, the partition function is dominated by state where $\lambda_{1\ldots\Bbbk} \sim \mathcal{O}(N)$, since in these states the non-singlet term in the Hamiltonian is suppressed. However, these are states in which the adjoints are infinitely far away, and the bulk theory is the same as in the singlet sector. Thus, we conjecture that the correct way to do this calculation is to first calculate the partition function at finite $N$ and then take the double-scaling limit in the final answer. In this order of limits, the dominance of the

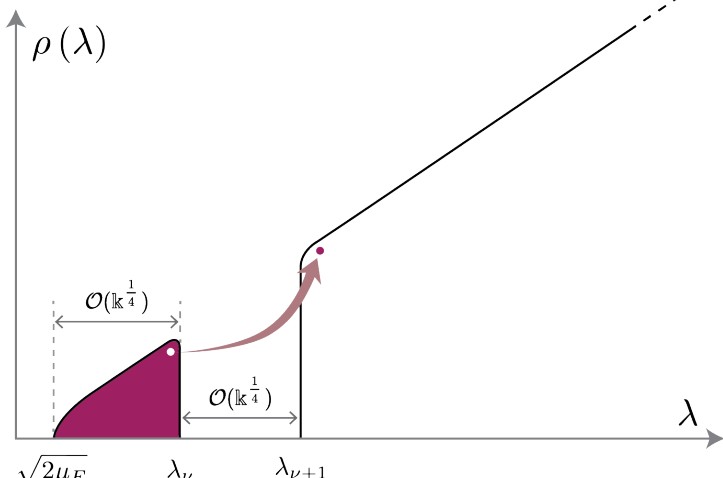

Figure 9: The evaporation process of the solid — an adjoint jumps out of the solid and into the rest of the Fermi sea.

hole-in-the-world state is clearer. It would be interesting to return to this question in future work.

### 4.3.1 Evaporation rate and page time

Let us now calculate the lifetime of the state above in the double-scaling limit $1 \ll \Bbbk \sim N^0$. We will find that it grows polynomially in $\Bbbk$, which, as we discuss in §5, we may consider to be a renormalised mass .

The action of an infinitesimal amount of time-evolution on the state yields just a phase with high probability, since it is an eigenstate, and with a small probability, a state where one adjoint escapes, i.e.,

$$e^{\iota H_{\mathrm{ns}}\epsilon}\,|\nu,\Bbbk\rangle = e^{\iota E_{\mathrm{ns}}\epsilon}\sqrt{1-\mathsf{r}}\,|\nu,\Bbbk\rangle + \iota\epsilon\sqrt{\mathsf{r}}\,|\nu,\Bbbk-1\rangle \otimes |\text{free adjoint}\rangle\,. \tag{123}$$

There is also a slight rearranging of the adjoints within the solid, as given by the $\alpha_j$ term in (112); the branch where the adjoint escapes is the one proportional to the $\beta_k$s in the same equation. We ignore the $\alpha_j$ branch for simplicity of exposition, but the following discusion is not modified by its inclusion.

The second branch in (123), where a single adjoint escapes, is illustrated in Figure 9. Because the hole-in-the-world state is an approximate eigenstate, the second branch has a much smaller amplitude that we have denoted by $\sqrt{\mathsf{r}}$. $\mathsf{r}$, which we may call an escape rate, can be estimated straightforwardly to be[21]

$$\mathsf{r} = \left\langle H_{\mathrm{ns}}^2\right\rangle - \langle H_{\mathrm{ns}}\rangle^2 \approx \Bbbk\,. \tag{124}$$

The log of this escape rate can also be thought of as the imaginary part of the energy of the state, and the lifetime of the state can be calculated using formulas for that of a resonance.

We adopt a more direct method here. Upon $n$ steps of time-evolution, we find

$$\left(e^{\iota H_{\mathrm{ns}}\epsilon}\right)^n |\nu,\Bbbk\rangle = e^{\iota E_{\mathrm{ns}}n\epsilon}(1-\mathsf{r})^{n/2}|\nu,\Bbbk\rangle + \iota n\epsilon\sqrt{\mathsf{r}}\,|\nu,\Bbbk-1\rangle \otimes |\text{free adjoint}\rangle + \mathcal{O}(\epsilon^2)\,. \tag{125}$$

---

[21]Consider a Hamiltonian $H = \begin{pmatrix} a & c \\ c^* & b \end{pmatrix}$. After time-evolution by an infinitesimal time $\epsilon$, the state $|0\rangle$ evolves to $e^{\iota H\epsilon}|0\rangle = e^{\iota a\epsilon}\left(1 - \frac{\epsilon^2}{2}|c|^2\right)|0\rangle + \iota\epsilon c\,|1\rangle$. Comparing with (123), we see that $\mathsf{r} = |c|^2$. It is also easy to verify that $\langle 0|H^2|0\rangle - \langle 0|H|0\rangle^2 = |c|^2$ in this two-state system.

Thus, we see that the branch where a single adjoint has escaped has an $\mathcal{O}(1)$ amplitude after time

$$t_{\text{esc}} \sim n\epsilon \sim \frac{1}{\sqrt{r}} \sim \frac{1}{\sqrt{\mathbb{k}}}\,. \tag{126}$$

The inverse of this quantity is the evaporation rate of the black hole.

Since there are $\mathbb{k}$ adjoints, the time for most of them to evaporate is

$$t_{\text{Page}} \approx \int_1^{\mathbb{k}} \mathrm{d}\mathbb{k}'\, t_{\text{esc}}(\mathbb{k}') \sim \sqrt{\mathbb{k}}\,. \tag{127}$$

We may consider this the Page time of this black hole.

The inclusion of a third branch in (123) corresponding to the $\alpha_j$s in (112) doesn't modify this discussion because it contributes to the variance at the same order, and so that rearrangement of the adjoint distribution becomes important on the same time-scale.

## 5 From the fixed-charge ensemble to the grand canonical ensemble

The partition function that [11] calculate isn't in the fixed-charge ensemble but in a grand canonical ensemble,

$$Z(\mathfrak{z}_b) = \int_{U(N)} \mathrm{d}\Omega\, e^{N\mathfrak{z}_b\left[\operatorname{tr}\Omega + \operatorname{tr}\Omega^\dagger\right]} Z(\Omega)\,, \tag{128}$$

where $Z(\Omega)$ is the partition function with twisted boundary conditions $X(\beta) = \Omega^\dagger X(0)\Omega$. In this section, we verify that our results are consistent with theirs and attempt to shed some light on the question of whether the $PU(N)$ symmetry is gauged or not.[22]

Let us first assume that the $PU(N)$ is a global symmetry and show that our results are consistent with those of [11, 12]. Since the twisted partition function is a class function [32], we can decompose it into fixed $PU(N)$ irrep sectors as

$$Z(\Omega) = \sum_r \frac{\chi_r(\Omega)}{d_r} Z_r\,. \tag{129}$$

Here, the characters are normalised so that

$$\chi_r(\mathbb{1}) = d_r\,, \qquad \int \mathrm{d}\Omega\, \chi_{r_1}(\Omega)^* \chi_{r_2}(\Omega) = \delta_{r_1, r_2}\,. \tag{130}$$

The factors $d_r$ of the dimension of the irreps ensures that

$$Z(\mathbb{1}) = \sum_r Z_r = \sum_r \operatorname{tr}_{\mathscr{H}_r} e^{-\beta H} = \operatorname{tr}_{\mathscr{H}}\left(\bigoplus_{\mathscr{H}_r} e^{-\beta H}\right) = \operatorname{tr}_{\mathscr{H}} e^{-\beta H}\,, \tag{131}$$

meaning that $Z(\mathbb{1})$ is the trace over the entire Hilbert space and $Z_r$ is the trace over the fixed-irrep Hilbert space $\mathscr{H}_r$. Here, unlike in some earlier section, $\mathscr{H}_r$ is not the irrep Hilbert space but the space of all states that transform in the irrep $r$ — in a hydrogen atom where the irreps are labelled by $l$, $\mathscr{H}_l$ is spanned by states with all $n, m$ indices consistent with $l$. This

---

[22]We thank Juan Maldacena for sharing some unpublished notes which directly inspired this section. Some of the statements were sharpened due to discussion during and after a talk at the Tata Institute of Fundamental Research, and we thank the audience there for the discussion.

justification for the factor of $d_r$ is only sensible in the case that $PU(N)$ is a global symmetry. We now show that the fixed irrep partition function we have found results in the partition function of [11] after the convolution (128) and a crucial renormalisation.

The first step is to expand the exponent

$$e^{N\mathfrak{z}_b\{\text{tr}\,\Omega+\text{tr}\,\Omega^\dagger\}} = \sum_{\mathbb{k},\mathbb{k}'=0}^{\infty} \frac{(N\mathfrak{z}_b)^{\mathbb{k}+\mathbb{k}'}}{(\mathbb{k}+\mathbb{k}')!}\binom{\mathbb{k}+\mathbb{k}'}{\mathbb{k}}(\text{tr}\,\Omega)^{\mathbb{k}}\left(\text{tr}\,\Omega^\dagger\right)^{\mathbb{k}'}. \tag{132}$$

Using the fact that $Z(\Omega)$ is independent of the global phase mode of $\Omega$ (by definition), we can integrate over it to find

$$\int_{U(1)_{\text{centre}}} e^{N\mathfrak{z}_b\{\text{tr}\,\Omega+\text{tr}\,\Omega^\dagger\}} = \sum_{\mathbb{k}=0}^{\infty} \frac{(N\mathfrak{z}_b)^{2\mathbb{k}}}{(\mathbb{k}!)^2}\left(\text{tr}\,\Omega'\right)^{\mathbb{k}}\left(\text{tr}\,\Omega'^\dagger\right)^{\mathbb{k}}, \tag{133}$$

where $\Omega'$ is valued in $PU(N)$. The integral over the phase mode is proportional to $\delta_{\mathbb{k},\mathbb{k}'}$. So we have, renaming $\Omega' \to \Omega$,

$$Z(\mathfrak{z}_b) = \sum_{\mathbb{k}} \frac{(N\mathfrak{z}_b)^{2\mathbb{k}}}{\mathbb{k}!^2} \int_{PU(N)} (\text{tr}\,\Omega)^{\mathbb{k}}\left(\text{tr}\,\Omega^\dagger\right)^{\mathbb{k}} \sum_r \frac{\chi_r(\Omega)}{d_r} Z_r. \tag{134}$$

The next step is to do the $\Omega$ integral

$$\frac{1}{d_{r_1} d_{\bar{r}_2}} \int_{U(N)} \chi_{f^{\otimes\mathbb{k}}\otimes\bar{f}^{\otimes\mathbb{k}}}(\Omega)\,\chi_{r_1\bar{r}_2}(\Omega). \tag{135}$$

Notice that we are integrating over all of $U(N)$ and not just $PU(N)$ — because the integrand is invariant and we aren't keeping track of normalisations, this makes no difference. We have also written the irrep $r$ as the symmetrisation of an irrep $r_1$ of the boxes and an irrep $\bar{r}_2$ of the anti-boxes. At leading order in large $N$, we can forget about the symmetrisation and take the full irrep to be simply $r_1 \otimes \bar{r}_2$.[23] Now, Schur-Weyl duality states that $\mathbb{k}$ copies of the fundamental decomposes into irreps with $\mathbb{k}$ boxes in the YD with multiplicity $d_r^{(\mathsf{S}_\mathbb{k})}$ equal to the dimension of the corresponding irrep of $\mathsf{S}_\mathbb{k}$. For the characters, this implies[24]

$$f^{\otimes\mathbb{k}} = \bigoplus_{r\,|\,\mathbb{k}} r_{U(N)} \otimes r_{\mathsf{S}_\mathbb{k}} \qquad \Rightarrow \qquad \chi_{f^{\otimes\mathbb{k}}} = \sum d_r^{\mathsf{S}_\mathbb{k}}\chi_r. \tag{136}$$

The symbol $r\,|\,\mathbb{k}$ denotes the restriction to representations with exactly $\mathbb{k}$ boxes. Thus, for $r_1, r_2$ with $\mathbb{k}$ boxes each, we find

$$\frac{1}{d_{r_1} d_{\bar{r}_2}} \int_{U(N)} \chi_{f^{\otimes\mathbb{k}}\otimes\bar{f}^{\otimes\mathbb{k}}}(\Omega)\chi_{r_1\bar{r}_2}(\Omega) \approx \frac{d_{r_1}^{\mathsf{S}_\mathbb{k}} d_{r_2}^{\mathsf{S}_\mathbb{k}}}{d_{r_1} d_{r_2}} \approx \binom{N}{\mathbb{k}}^{-2} \approx \frac{(\mathbb{k}!)^2}{N^{2\mathbb{k}}}. \tag{137}$$

Here, we've used (130) and (136) in the first equality and (29) in the second equality. The first equality is only approximately true because we're ignoring the possibility that some pairs of fundamentals and anti-fundamentals could fuse to a singlet state.

The final partition function, plugging (137) into (134) and using the KT free energy, is

$$Z(\mathfrak{z}) = \sum_{\mathbb{k}} \mathfrak{z}_b^{2\mathbb{k}} \sum_{r_1,r_2\,|\,\mathbb{k}} Z_{r_1\bar{r}_2} = \sum_{\mathbb{k}} \mathfrak{z}_b^{2\mathbb{k}}\left(\frac{N}{\sqrt{\mathbb{k}}}\right)^{(2-R)\mathbb{k}} = \sum_{\mathbb{k}}\left(\mathfrak{z}_b N^{\frac{2-R}{2}}\right)^{2\mathbb{k}}\left(\sqrt{\mathbb{k}}\right)^{-(2-R)\mathbb{k}}. \tag{138}$$

---

[23]At finite $N$, $r_1 \otimes \bar{r}_2$ contains irreps with fewer than $\mathbb{k}$ boxes as well.

[24]The fact that each irrep appears with such a multiplicity can be rephrased as the statement that we are summing over irreps with a 'Poissonised-Plancherel' measure [19].

The quantity within the brackets is exactly the renormalised fugacity (17).[25] Evaluating the sum by saddle-point, we find exactly the answer (18) from [11] with $\mathfrak{z}$ being the *renormalised* fugacity.

Let us observe that we can define a renormalised fixed-charge partition function[26]

$$Z_{\mathrm{ren},\Bbbk} = \left(\sqrt{\Bbbk}\right)^{(2+R)\Bbbk}, \qquad S_{\mathrm{ren}} = 2\Bbbk \log \sqrt{\Bbbk}, \qquad E_{\mathrm{ren}} = -\frac{\Bbbk}{2\pi} \log \sqrt{\Bbbk}, \tag{139}$$

which is dual to the renormalised fugacity $\mathfrak{z}$ rather than the bare fugacity $\mathfrak{z}_b$,

$$Z(\mathfrak{z}) = \sum_{\Bbbk} \frac{\mathfrak{z}^{2\Bbbk}}{\Bbbk!^2} Z_{\mathrm{ren},\Bbbk}. \tag{140}$$

Comparing this with (134), it seems that $Z_{\mathrm{ren},\Bbbk}$ is a renormalised version of the fixed $\Bbbk$ partition function $\int_{PU(N)} (\mathrm{tr}\,\Omega)^{\Bbbk} \left(\mathrm{tr}\,\Omega^{\dagger}\right)^{\Bbbk} Z(\Omega)$. We will return to this point in §5.1 and §6.4.

## 5.1 Gauging as renormalisation

We finally address the difference between our approach and that of [16], which is the question of whether the $PU(N)$ symmetry should be gauged or not. We find that, even if we start with the ungauged theory, a gauged description might still emerge after renormalisation in the double-scaling limit.

To begin, we note that we can rewrite the twisted partition function $Z(\Omega)$ as the path integral of a gauged theory,[27]

$$Z(\Omega) = \int_{Pe^{i\oint A} = \Omega} \frac{[\mathscr{D}X][\mathscr{D}A]'}{\mathrm{vol}\,(\mathrm{gauge})} e^{-S[X,A]}. \tag{141}$$

The integral over $A$ excludes the zero-mode, which is fixed. This is the same as the ungauged twisted partition function, since for every $A$ degree of freedom, i.e., the non-zero modes of $A$, that we include in $[\mathscr{D}A]$, we introduce a corresponding constraint. The zero-mode of the gauge field is just the twist.

Where the gauged and ungauged partition function differ is in the definition of the fixed-irrep partition function. In the ungauged theory, it can naturally be defined as the trace over a Hilbert space as in (131). In the gauged theory, however, the most sensible definition is

$$Z_r^{(g)} = \int \frac{[\mathscr{D}X][\mathscr{D}A]}{\mathrm{vol}\,(\mathrm{gauge})} \mathrm{tr}_r \left(Pe^{i\oint A}\right) e^{-S[X,A]}. \tag{142}$$

We have reintroduced the integral over the zero-mode of $A$ here. This was how the adjoint partition function was defined in [16]. Comparing this with (129), we find that

$$Z_r^{(g)} = \frac{Z_r}{d_r} \xrightarrow{\text{dominant YD}} e^{-R\Bbbk \log \frac{N}{\sqrt{\Bbbk}}}. \tag{143}$$

The partition function of [11,12] is then

$$Z(\mathfrak{z}_b) = \sum_{\Bbbk=0}^{\infty} \frac{(N\mathfrak{z}_b)^{2\Bbbk}}{\Bbbk!^2} \int \mathrm{d}\Omega\; \chi_{f^{\otimes\Bbbk} \otimes \bar{f}^{\otimes\Bbbk}}(\Omega) \sum_{r_1,r_2} \chi_{r_1 \bar{r}_2}(\Omega)\; e^{-R\Bbbk \log \frac{N}{\sqrt{\Bbbk}}}. \tag{144}$$

---

[25]We thank Juan Maldacena for pointing this out.

[26]This quantity was originally defined by Juan Maldacena in unpublished notes.

[27]We thank Shiraz Minwalla for emphasising this fact to us.

Table 1: The entropy of different ensembles in the gauged and ungauged models.

| | Entropy | Source of Entropy | Discussion in paper |
|---|---|---|---|
| Ungauged MQM | | | |
| Fixed irrep $r$ | $\log d_r \approx 2\Bbbk \log \frac{N}{\sqrt{\Bbbk}}$ | Degeneracy due to symmetry | §3 |
| Fixed charge $\Bbbk$ | $2\Bbbk \log N$ | Dimension of dominant irrep and its multiplicity | Around (136) |
| Gauged MQM | | | |
| Fixed irrep $r$ | $\mathcal{O}(\log \Bbbk)$ | Fluctuations of the hole-in-the-world state | Above (119) |
| Fixed charge $\Bbbk$ | $\Bbbk \log \Bbbk$ | Multiplicity of dominant irrep | Around (145) |

The $\Omega$ integral gives, as in (137),

$$\sum_{r_1,r_2} \int d\Omega \, \chi_{f^{\otimes \Bbbk} \otimes \bar{f}^{\otimes \Bbbk}}(\Omega) \, \chi_{r_1 \bar{r}_2}(\Omega) \approx \sum_{r_1} d_{r_1}^{(S_\Bbbk)} \sum_{r_2} d_{r_2}^{(S_\Bbbk)} \approx \Bbbk! \, . \tag{145}$$

The sums are done as in §3.2, and we know from there that they are dominated by one irrep. Thus, we see that the fixed $\Bbbk$ partition function has extra entropy of $2\Bbbk \log \sqrt{\Bbbk}$ compared to the fixed-irrep partition function, and that this entropy is entirely from the multiplicity of the dominant irrep in the product of $\Bbbk$ adjoints.

Absorbing the powers of $N$ to renormalise $\mathfrak{z}$, we see that the fixed-$\Bbbk$ partition function in the gauged interpretation is

$$Z_\Bbbk^{(g)} = \left(\sqrt{\Bbbk}\right)^{(2+R)\Bbbk} = Z_{\mathrm{ren},\Bbbk} \, , \tag{146}$$

i.e. we get exactly the renormalised partition function defined in (139).

So we see that we can reproduce the result of [11, 12] from both the gauged as well as the ungauged interpretations. Since the gauged entropy agrees with the renormalised entropy defined in (139), we speculate that the passage from the ungauged theory to the gauged one corresponds to a renormalisation in the double-scaling limit. We give pictorial evidence for this speculation in §6.3. We leave further analysis of this difference to future work.

## 5.2 Entropy of the black hole

Now, for the sake of clarity, we collect our results on the entropy. There are three important partition functions: the fixed-irrep partition function, the fixed-charge partition function, and the grand canonical partition function. We summarise the main points in table 1.

There are two versions of the fixed-irrep partition function: the gauged and the ungauged. The ungauged model has an entropy given by the log of the dimension of the irrep dimension, as we found in §3; for the dominant irrep at a given $\Bbbk$ this entropy is $2\Bbbk \log N/\sqrt{\Bbbk} + \mathcal{O}(\Bbbk)$. The gauged fixed-irrep partition function (142), which is really the expectation value of a Wilson line wrapping the thermal circle, is the number of excitations of the hole-in-the-world state that change the energy only at $\mathcal{O}(\Bbbk)$. As discussed above (119), this entropy is $\mathcal{O}(\log \Bbbk)$.[28]

For the fixed-charge partition function, there is a new contribution to the entropy from the dominant irrep appearing with a certain multiplicity in the product of $\Bbbk$ adjoints. This contribution is $\Bbbk \log \Bbbk$, as we discuss around (145). The grand canonical ensemble is at leading order equivalent to the fixed-$\Bbbk$ ensemble and so there is nothing new in this case.

As we have argued in sections 5.1 and 6.3, the ungauged entropy is dominated by IR degrees of freedom that decouple from the bulk theory and thus shouldn't count towards the entropy of the black hole. The actual entropy of the black hole should be the log of the number of states in the gauged theory in the sector with $\Bbbk$ adjoints,

$$S_{\mathrm{BH}} = \Bbbk \log \Bbbk + \mathcal{O}(\Bbbk) \, . \tag{147}$$

---

[28]This is in agreement with unpublished results of Juan Maldacena.

We prefer this answer over the number of states in the dominant irrep because to actually see this state we would need to collapse $\Bbbk$ adjoints; we would see $e^S$ distinct bound states, corresponding to the different permutations of the $\Bbbk$ adjoints that we collapsed.

# 6 Discussion

In this work, we have found a Hamiltonian derivation of the free energy of the two-dimensional black hole. The microstates of the black hole are dual to the subspace of $\Bbbk$ adjoints of the $PU(N)$ symmetry. The entropy can be calculated either directly in terms of $\Bbbk$ indistinguishable adjoints or by counting the dimension of the irrep that dominates this tensor product. The energy was calculated by a direct analysis of the Schrödinger equation, and finding a non-trivial bound state that dominates the fixed-charge partition function. Since [13] had two interpretations that resulted in different free energies, the final outcome is not only that we have *reproduced* the free energy but that we have *picked out* one of these two prescriptions.

Our main result, however, is the existence of a novel, long-lived bound state — the hole-in-the-world state — that has properties very reminiscent of a black hole. We will discuss further the ways in which the properties of this state match the two-dimensional bulk description in §6.1, but we may also wonder about higher-dimensional black holes. The purely two-dimensional black hole has the unsatisfactory property that, because $\Bbbk-2$ is small, strings on this background are strongly coupled, because of which its black hole-like nature is unclear. However, as discussed recently in [50] among others, the $SL(2,\mathbb{R})_{\Bbbk}/U(1)$ CFT when considered as a two-dimensional sector of a higher-dimensional string theory can have $\Bbbk > 3$ where it is weakly coupled. Further, the WZW model is believed to be exactly dual to a sine-Liouville CFT at arbitrary $\Bbbk$ and so sine-Liouville theory is relevant for the description of higher-dimensional black holes. While we have not verified that the MQM description is relevant in this higher-dimensional setting, our result indicates that the Lorentzian description of the microstates in the sine-Liouville CFT might be a condensate of stretched strings similar to our hole-in-the-world state. In other words, we speculate that this hole-in-the-world state points the way to a more general Lorentzian description of the winding condensate that is believed to carry all the entropy of a black hole, see e.g. [51].

Another important avenue to explore in the future is to obtain the stringy counterparts of the black hole microstates we found in this paper on the *bulk* side, specifically on the cigar, using the off-shell formulation of string theory presented in [52, 53].

In the rest of this section, we speculate on what our results might imply for the string-theoretic interpretation of this system. First, in §6.1 we discuss aspects of our central result — the hole-in-the-world state. We heuristically sketch out ways in which it behaves like a black hole and the sine-Liouville string theory. We go on in §6.2 with a short summary of the thermal phase structure of string theory, and what our results seem to imply for the phase structure of the $c = 1$ theory. In §6.3, we present a natural extension of the string theory construction of [16] that seems consistent with our results; it also provides some motivation for thinking of the passage from the ungauged MQM to the gauged one as renormalisation in the double-scaling limit. Finally, in §6.4, we point out that our count of states is related to a random walk model called the Motzkin walk model; we speculate on the interpretation of this in string-theoretic terms.

## 6.1 Bulk properties of the hole-in-the-world state

### 6.1.1 Stringy justification for the bound state (or) how we learned to stop worrying and love the hole in the world

Since the spatial direction of the string theory target space is identified with the Fermi surface, one might worry that a hole in the eigenvalue distribution shouldn't be allowed because it would cause a radical change on the string theory side. In this section, we argue heuristically that the formation of the hole is dual to the passage from sine-Gordon theory to sine-Liouville theory in the bulk.

In the usual duality between the singlet sector of the MQM and $c = 1$ string theory, a probe closed string is dual to a particle-hole pair in the Fermi sea. In the bulk, such a string scatters off the tachyon wall — the $\mu e^{-2\phi}$ term in the worldsheet action — and in the MQM the particle-hole pair scatters off the edge of the Fermi sea at $\lambda = \sqrt{2\mu_F}$. Thus, the reflection amplitudes in both the bulk and the boundary are non-analytic in $\mu$.

In the hole-in-the-world state, a particle-hole pair now scatters off the new edge of the Fermi sea at $\Bbbk^{1/4} \sim \mathfrak{z}^{\frac{1}{2-R}}$ — note that the $\mathfrak{z}$ scaling is that found in (22). In sine-Gordon theory, the worldsheet action contains a term $\left(\mathfrak{z}^{\frac{2}{2-R}} e^{-2\phi}\right)^{\#}$, much as it contains a term $\mu e^{-2\phi}$. We see that the coefficient of $e^{-2\phi}$ is exactly the square of the position of the edge of the Fermi sea in both cases. Thus, it is plausible that scattering of closed strings at large $\mathfrak{z}$ and the scattering of particle-hole pairs in the hole state are non-analytic in the same parameter $\mathfrak{z}^{\frac{2}{2-R}}$ and are further exactly dual. For example, the leading non-analyticity of the susceptibility at large $\mathfrak{z}$ calculated in [12] is precisely $\log \mathfrak{z}^{\frac{2}{2-R}}$. Such similarities between $\mu$ and $\mathfrak{z}^{\frac{2}{2-R}}$ can also be seen in the reflection coefficients, see e.g. [54, 55]. We leave a detailed check of such a duality to future work.

More evidence of the duality with sine-Liouville can be seen in the excitations of the solid itself. Since it is a compact region in the double-scaling limit, we expect that the spectrum of these excitations is quantised. Further, we should not be able to excite the eigenvalues without 'carrying around' the adjoints, since the adjoint index is associated with the corresponding eigenvalue.

Sine-Liouville theory *also* has a spectrum of bound states in which momentum and winding modes are excited together,

$$j_N = \frac{\Bbbk|w| - |n|}{2} - N \in \left(\frac{1}{2}, \frac{\Bbbk-1}{2}\right), \qquad N \in \mathbb{N},$$

$$h_{jnw} = -\frac{j(j-1)}{\Bbbk-2} + \frac{(n-\Bbbk w)^2}{4\Bbbk}, \qquad \bar{h}_{jnw} = -\frac{j(j-1)}{\Bbbk-2} + \frac{(n+\Bbbk w)^2}{4\Bbbk}. \tag{148}$$

Here, $n$ counts momentum modes and $w$ winding modes. The constraint on $j_N$ ensures that momentum modes can't be excited without winding modes. We can associate a momentum mode with the eigenvalue and the winding mode with an adjoint. Thus, the constraint is entirely analogous to what we expect from the solid.

Checking this duality by making the preceding statements precise is a promising avenue of future work.

### 6.1.2 The hole-in-the-world and black holes

The hole-in-the-world state bears several qualitative resemblances to black holes. The state radiates into free particles — because the interaction between adjoints falls off like $1/(\Delta\lambda)^2$, an adjoint far away from the solid will behave to first order as a free particle.

As discussed in §4.3.1, the state radiates like a black hole at a slow rate, with a lifetime polynomially large in $\Bbbk$. When considering the bound state in $\tau$-space, which is perhaps more natu-

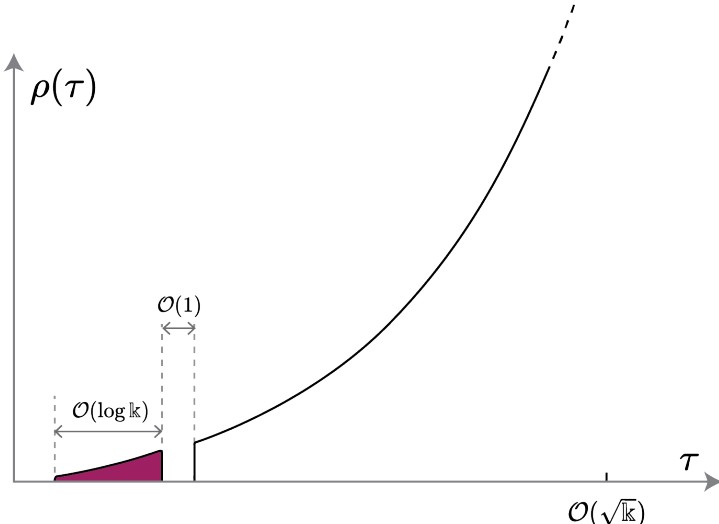

Figure 10: A depiction of the eigenvalue distribution shown in Fig. 8, this time with respect to $\tau \sim \log \lambda$. We note that in $\tau$-space, the hole-in-the-world's size does not scale with $\Bbbk$. The disconnected solid of non-singlets is of size $\mathcal{O}(\Bbbk)$, and its Schwarzschild radius is of size $\mathcal{O}(\Bbbk)$.

ral from the perspective of the free adjoints, the size of the hole is always $\mathcal{O}(\lambda_{\nu+1}/\lambda_\nu) = \mathcal{O}(1)$, while the size of the bound solid itself is $\mathcal{O}(\log \Bbbk)$. We can see from the discussion in §4.2.1 that a configuration of free adjoints will only be energetically favored when they are spread over a region $\Delta \lambda$ of size $\log \Delta \lambda \sim \Delta \tau \sim \mathcal{O}(\Bbbk^{1/2})$, which we can interpret as a Schwarzschild radius. We note that both the evaporation time and this effective radius are polynomial in the mass of the black hole. This distribution of scales in $\tau$-space is shown in Fig. 10:

This radiation can furthermore be mined as the black hole radiates entirely. It would be interesting to compare the resulting entanglement curve between the solid and the collected radiation with the semiclassical Liouville calculation [56].

The bound state of eigenvalues further bears resemblance to bound $D$0-brane models of black holes in BFSS, whose evaporation processes were considered in [57]. However, in such models the bound states radiate due to a supersymmetry-induced cancellation of strong attractive forces due to open strings stretching between the branes. Despite being non-supersymmetric, this model reproduces similar behavior.

## 6.2 Thermal phase transitions in string theory

In this subsection, we begin with a review of some of the classic results about the canonical ensemble in string theory [36,37,58] and try to put the story reviewed in §2 into the context of this literature. Finally, we attempt to physically interpret the fact that our calculation picked out the second prescription of [13].

One of the first conclusions one can reach from the spectrum of closed string states is that there's a Hagedorn divergence in the one-loop partition function. This is because the log of the density of states has the same asymptotic behaviour as the energy spectrum at large dimension — both grow as $\sqrt{\Delta}$ where $\Delta$ is the conformal dimension of the oscillators (that of the full vertex operator must be 2). As a result, the partition function looks like

$$Z(\beta) \sim \int \mathrm{d}E \, e^{\beta_H E} e^{-\beta E}, \tag{149}$$

Worldsheet ($S^2$ with vortices)

Target spacetime

Figure 11: A vortex is a hole that winds around the thermal circle.

where the first piece is the density of states and $\beta_H = 4\pi\sqrt{\alpha'}$ is the reciprocal of the Hagedorn temperature $T_H$; at $T > T_H$ the exponent grows at large energies and the integral diverges. Naïvely then, it appears as if string theory has a limiting temperature.

Further progress was made by [36, 37], who made a remarkable observation about this temperature. The observation was that $T_H$ is exactly the temperature at which two things happen:

1. The lowest *winding mode* of the string becomes massless. In the expansion of the worldsheet field $T(\tau, \sigma)$ where $T$ is the target space Euclidean time and $\tau, \sigma$ are worldsheet coordinates,

$$T = T_0 + \frac{n}{R}\tau + mR\sigma + \text{oscillators},\tag{150}$$

the mode with no oscillators excited and $n = 0, m = 1$ is the lowest winding mode, since it is the lowest mass-squared mode where a spatial circle of the string winds around the cylinder. The corresponding vertex operator is $e^{\iota R(T_L - T_R)}$. Its mass depends on $R$; at lower temperatures, it has $m^2 > 0$ and at higher temperatures it has $m^2 < 0$.

This means that this mode becomes a *winding tachyon* at $T > T_H$ and so one should think of the Hagedorn temperature as the location of a phase transition where this tachyon condenses.

Of course, there is the question of whether the winding tachyon potential has a stable minimum.

2. The worldsheet undergoes a Berezinskii-Kousterlitz-Thouless (BKT) transition, see e.g. [59] for an introduction, in which vortices get deconfined and proliferate.

From the stringy point-of-view, a vortex is just a hole that winds around the thermal circle, see figure 11. Interestingly enough, since a vortex needs to have a core where the derivatives diverge, this proliferation also introduces a *worldsheet UV cutoff* — meaning that conformal invariance is broken on the worldsheet. Since conformal invariance was the main reason that the $S^2$ partition function of string theory vanished, this phase transition also causes the genus 0 partition function[29] to become non-zero.

The fact that these happen at the same temperature is not terribly surprising, since the insertion of a winding mode operator can equivalently be thought of as that of a vortex.[30] The condensation of the winding tachyon, then, is just the proliferation of vortices.

---

[29]As a pedantic point, we note that introducing holes causes the Euler characteristic to change but not the genus.

[30]To see this, consider a small disc around a winding mode operator. Using the state-operator correspondence and the fact that the state looks like (150) with $n = 0, m = 1$, the boundary of this disc winds around the thermal circle. This is the same property that characterises a vortex.

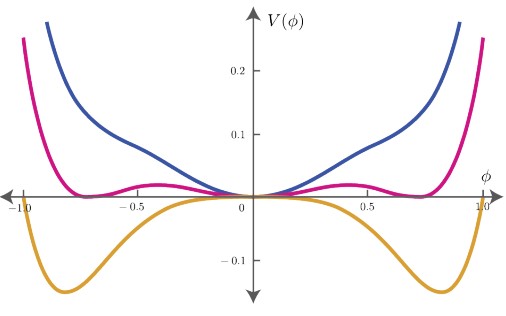

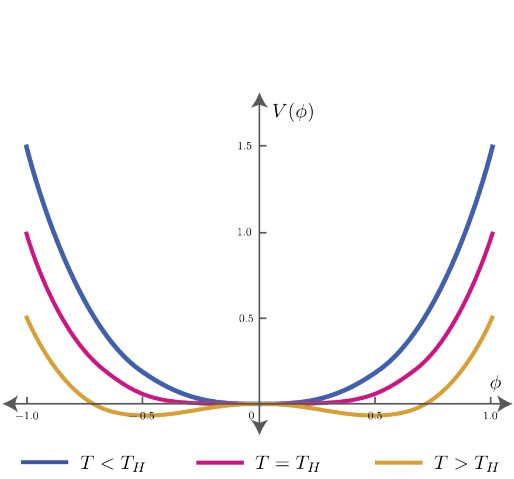

(a) The Landau-Ginzburg potentials $m^2\phi^2 + u\phi^4$ for $u > 0$. There is only one minimum for positive $m^2$ ($T < T_H$) and two minima at negative $m^2$ ($T > T_H$) with a smooth transition.

(b) The potential $m^2\phi^2 - u\phi^4 + v\phi^6$ for $u, v > 0$. The non-trivial minimum becomes the true vacuum at positive $m^2$ ($T < T_H$). The bottom-most line is the $m^2 = 0$ ($T = T_H$) graph; we can see that there is nothing special about the erstwhile Hagedorn temperature in this model. [58] argue that this is the model that more closely describes string theory in flat space, except that the transition is to an unstable phase — we have chosen to make the new phase stable in this cartoon for simplicity.

Figure 12: A cartoon of how a first-order transition happens at a lower than expected temperature. In both cases, increasing the temperature takes $m^2$ from positive to negative.

Despite their equivalence, the BKT point of view has an important advantage: the question of whether the winding tachyon potential has a stable minimum becomes the question of whether the worldsheet theory above the BKT transition flows to a non-trivial CFT when you add a fugacity for the winding mode operator, see figure 1 and subsequent discussion in [36] for an exceptionally clear exposition. In $\mathbb{R}^{25} \times S^1$, the answer is no; there is a Jeans instability where, because of the infinite volume of space, the matter starts "clumping" together.

In the $c = 1$ case, however, the answer is yes — the sine-Liouville theory is a non-trivial CFT at a non-zero value of the fugacity $\mathfrak{z}$ for $R < 2$. An important point here is that the sine-Liouville theory cannot be studied perturbatively in $\mathfrak{z}$, see the discussion around (12), which can be thought of as the fact that adding this term triggers a non-trivial RG flow to a new fixed point. This new fixed point, in turn, can be thought of as being dual to our hole-in-the-world state. The fact that the arguments of [36,37] predict the location of the transition at $R = 2\sqrt{\alpha'}$ in the $c = 1$ model was noted in [38].

An important point added by [58] was about the order of the transition. Since the mass-squared of the winding mode goes smoothly to 0 and then negative values as temperature is increased, the above discussion might lead one to conclude that the Hagedorn transition is second or higher order.[31] However, it turns out that the coupling between the winding mode and the dilaton modifies the effective Landau-Ginzburg potential and the transition is first order and, more importantly, *it occurs at a lower temperature $T_c < T_H$*. A simple cartoon for this can be seen in figure 12.

How do the two interpretations mentioned in §2 tie in to this discussion? Firstly, both interpretations agree on the fact that there is an actual transition of $c = 1$ string theory at the Hagedorn value $R = 2$.[32] This is similar to figure 12a. The first interpretation, however,

---

[31]It should be noted that BKT transitions are generically of infinite order.

[32]See [11] for a discussion of this transition in the context of the first interpretation.

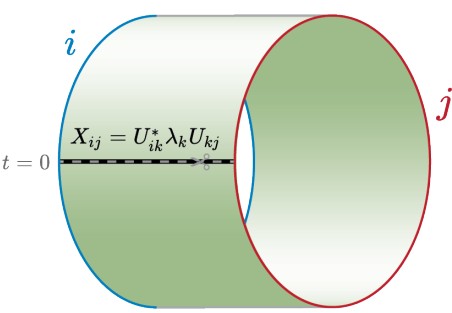

(a) The simplest Feynman diagram with one vortex (blue) and one anti-vortex (red). The point ($t = 0$) on the thermal circle where we cut the diagram has been indicated by a dashed line. The corresponding propagator $X_{ij}$ has been shown in black.

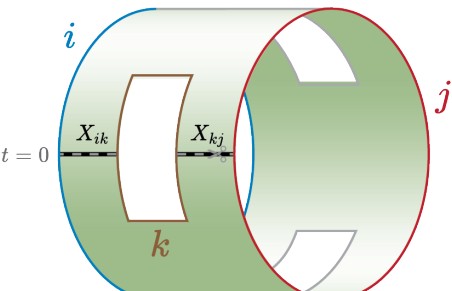

(b) A slightly more complicated Feynman diagram with one vortex and one anti-vortex. It has topologically trivial faces that don't wrap the thermal circle. The face in the middle carries the index $k$ and the propagators adjoining it correspond to the matrix elements $X_{ik}$ and $X_{kj}$ respectively.

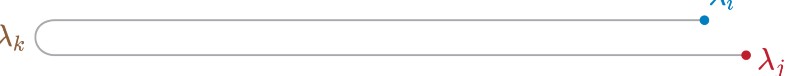

(c) Our speculation for the stretched string interpretation of the previous Feynman diagrams at $t = 0$.

Figure 13: Feynman diagrams with one vortex and one anti-vortex.

posits a *second* Hagedorn transition at the black hole temperature $R = 3/2$ and that is why the sphere free energy vanishes [11]. The free energy in the second interpretation is completely analytic at the black hole temperature and so under this interpretation there is only one phase transition for $R \geq 3/2$.

Thus, our results imply that two-dimensional string theory has only one Hagedorn-type phase transition, at the naïve Hagedorn temperature. Various arguments suggest another phase transition at $R = 1$, and our results have no bearing on this question. The entropy calculation presented in Appendix A is likely relevant for this highest temperature phase.

## 6.3 A stringy picture for the entropy

In this section, we attempt to understand the relation between the MQM and the stretched strings of [16]. We will also find a natural picture for the renormalisation from the ungauged MQM to the gauged one we noted in §5.1. This section is entirely speculative and we hope to return to some of these ideas in more detail in future work.

We begin by going to the deep UV of the worldsheet, which is a Feynman diagram for the MQM, as emphasised in §2. A Feynman diagram that corresponds to the genus 0 partition function in the adjoint sector is one with the topology of $S^2$ and with two faces that wind around the thermal circle in opposite ways. These two faces are the vortex and the anti-vortex respectively, see figure 14a. For simplicity, we consider the case where the vortices are the only non-trivial faces. The vortex and the anti-vortex have an index each, which we call $i, j$ respectively.

This is a cylinder diagram corresponding to the propagation of the matrix element $X_{ij}$ around the circle. Using the eigenvalue decomposition, this matrix element can be written as $\left(U^\dagger\right)_{ik} \lambda_k U_{kj}$. The $i, j$ indices are the right-action indices that carry the $PU(N)$ symmetry and the $k$ index is the left-action index whose wavefunction is given by the eigenvalue equation. In the double-scaling limit, $k$ is kept at $\mathcal{O}(1)$ inside the solid, while $i, j$ are unconstrained and

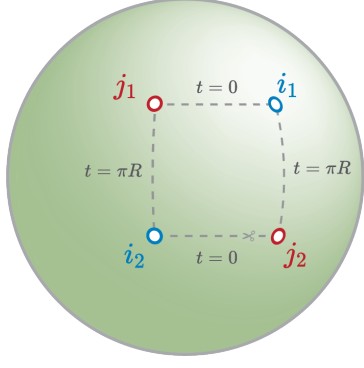

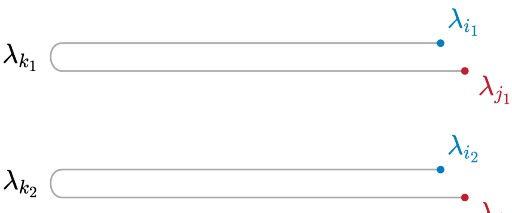

(a) A simple Feynman diagram with two vortices and two anti-vortices. It has two faces with no winding (one not visible), two with winding 1 and two with winding −1.

(b) A stretched string interpretation of the Feynman diagram to the left, at $t = 0$. There is a similar situation on the other side of the thermofield double, but the strings stretch between different pairs of eigenvalues there as can be read off from the dotted lines in figure 14a.

Figure 14: Feynman diagrams with two vortices and two anti-vortices.

therefore typically of $\mathcal{O}(N)$.

Now, to give this Feynman diagram a Hamiltonian interpretation, we cut it at $t = 0$. Remembering that the eigenvalue corresponds to the spatial direction in the string theory, we can associate the cut loop at $t = 0$ to a string that goes from $\lambda_i$ to $\lambda_k$ and comes back to $\lambda_j$, see figure 13c. In the double-scaling limit, this is an open string that comes in from $\infty$ to $\lambda_k$ and goes back out — similar to the stretched string of [16].

In [16], the stretched string is anchored to an FZZT brane at $\infty$. However, in our picture, the open string has two endpoints at different eigenvalues $\lambda_i, \lambda_j$, which one might naively associate to two different branes. One way to make our picture consistent with that of [16] would be to remember that $i, j$ are summed over all $\mathcal{O}(N)$ values, which could be consistent with the fact that an FZZT brane ranges over all values of the dilaton from some $\phi_{\min}$ to $\infty$. We caution the reader that this is a departure from the precise solution in [16], were the two ends of the stretched string were at the same dilaton value (which is the same eigenvalue, up to an integral transformation). Since the entropy count was a sum over the different values that $i, j$ could take, this suggests that the stringy picture for the entropy is related to the length of the FZZT brane. The fact that the latter diverges is nothing but the fact that the entropy depends on $N$ and also diverges in the double-scaling limit.

Before moving on to the $\Bbbk = 2$ case with two vortices, let us address how to think about this when there are $n_F$ topologically trivial faces between the vortex and the anti-vortex, as in figure 13b. In this case the matrix we have to diagonalise is $(X_1)_{ii_1}(X_2)_{i_2 i_3}...(X_{n_F+1})_{i_{n_F} i}$. The rest of the discussion carries through at the low level of precision we have maintained in this section.

Let us now look at the case of two vortices, which we can draw as in figure 14a. At $t = 0$ we see two stretched strings; and two corresponding stretched strings at $t = \pi R$. The entire discussion carries through for each string, with the small caveat that we can exchange the two strings by shuffling the indices on the vortices. Therefore the number of states passing through $t = 0$ is a factor of 2 less than what one might expect from two stretched strings.

More generally, at arbitrary $\Bbbk$, each stretched string should be thought of as an adjoint, and the different stretched strings as indistinguishable. The number of states is therefore counted by the number of states in $\Bbbk$ indistinguishable copies of the adjoint — which was exactly the counting we did in §3.1. Thus, the sine-Liouville model seems to directly realise one of the two countings we have presented. We hope to return to these ideas and make them precise in future work.

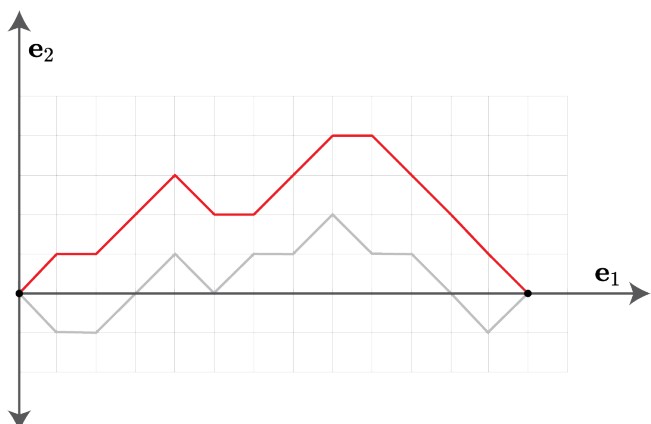

Figure 15: A valid example (red) and an invalid example (gray) of an uncoloured Motzkin walk.

Further, the renormalised partition function (139), which we saw in §5.1 to be natural for the gauged theory, has a natural interpretation here. This is because the entropy we calculated in §3 corresponds to the number of places the open string can end, and the indices at the edge of the open strings are infinitely far away in the double-scaling limit. These indices thus shouldn't matter very much for dynamics at finite distance, and so the renormalisation associated to the double-scaling limit could be regarded as decoupling the $PU(N)$ global symmetry from the bulk dynamics by sending the objects charged under the symmetry (the ends of the open strings) infinitely far away.

Finally, the major caveat about the pictures in this section is that, in this interpretation, the stretched string seems to 'jump' across the hole in the world. This suggests that there might be a more useful picture in the black hole phase. This is what we turn to now.

## 6.4 Motzkin walks

We end our discussion by describing another equivalent way of performing the entropy calculation in 3.2 using a class of discrete random walks called *Motzkin walks*. As we will explain, this allows us to draw a speculative but tantalizing connection between the solutions of the cigar string theory and these random walks.

First, some definitions. Let $\{\mathbf{e}_1, \mathbf{e}_2\}$ be the standard basis of $\mathbb{R}^2$. A Motzkin walk (or path) of length $n$ is a continuous lattice path from $(0, 0)$ to $(n, 0)$ that never dips below the $x$-axis created out of the following unit steps:

$$\{\mathbf{e}_1\} \cup \{\mathbf{e}_1 + \mathbf{e}_2, \mathbf{e}_1 - \mathbf{e}_2\}. \tag{151}$$

For obvious reasons, the above are often referred to as 'flat' steps, 'up' steps and 'down' steps respectively. An example of a valid and an invalid Motzkin walks are illustrated in figure 15. The number of Motzkin walks of length $n$ are counted by the $n^{\text{th}}$ Motzkin number, $\mathscr{T}_n$. These numbers appear while enumerating a host of other combinatorial objects — amongst other things, $\mathscr{T}_n$ counts the number of non-intersecting chords that can be drawn between $n$ points on a circle, the number of grammatically allowed ways of placing left and right parentheses, and spaces in a sentence and the number of configurations of a spin-1 chain satisfying $\sum_{i=1}^{m} S_i^z \geq 0$ for $m < n$ and $\sum_{i=1}^{n} S^z = 0$. Given their prevalence then, it is no surprise that these numbers have been quite extensively studied in the combinatorics literature. For a more complete list of the different manifestations of Motzkin numbers and a thorough discussion of their properties and relations, see [60].

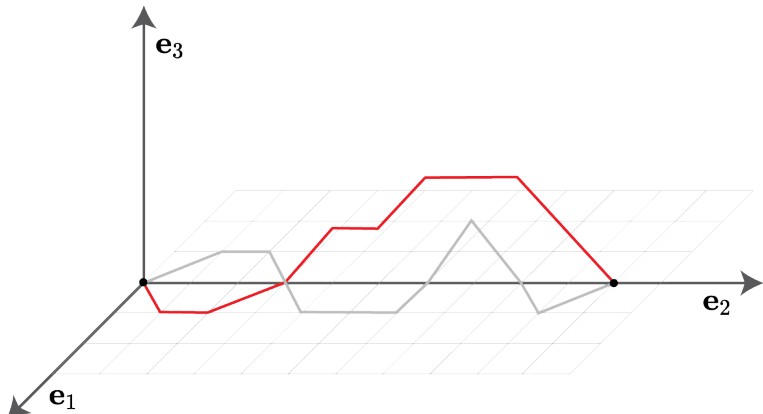

Figure 16: A valid (red) and an invalid (gray) example of a 2-coloured Motzkin walk.

In what follows, we will be interested in a higher dimensional generalization of the Motzkin walk — for historical reasons, these extra dimensions are often referred to as *colours*, and these generalized walks are called *coloured* Motzkin walks [61].[33] Once again, let $\{\mathbf{e}_1, \ldots \mathbf{e}_s, \mathbf{e}_{s+1}\}$ be the standard basis of $\mathbb{R}^{s+1}$. An $s$-coloured Motzkin walk is a lattice path from $(0, \ldots, 0)$ to $(n, 0, \ldots, 0)$ that always remains in the positive 'quadrant' of $\mathbb{R}^{s+1}$, created out of the following unit steps,

$$\{\mathbf{e}_1\} \cup \{\mathbf{e}_1 + \mathbf{e}_2, \, \mathbf{e}_1 - \mathbf{e}_2\} \cup \left( \bigcup_{i=2}^{s} \{\mathbf{e}_1 - \mathbf{e}_i + \mathbf{e}_{i+1}, \, \mathbf{e}_1 + \mathbf{e}_i - \mathbf{e}_{i+1}\} \right). \tag{152}$$

An example of a valid and an invalid 2-coloured Motzkin walks with ten steps are illustrated in figure 16. We will denote the number of Motzkin walks of length $n$ with *at most $s$* colours by $\mathscr{T}_n^s$. The constraint that the walker stay in the positive quadrant bounds the maximum number of colours $s$ by half the number of steps $n/2$.

The relevance of these walks to our story begins with the highly non-trivial observation that the number of sYTs with $n$ boxes and at most $2s + 1$ rows is also given by $\mathscr{T}_n^s$.[34] Indeed, the authors of [61] take this a step further and construct a bijection between the two sets, assigning each Motzkin walk to a specific sYT. The complete bijection is rather involved and will not be necessary for our present discussion. However, the mere existence of such a map implies that we can reproduce everything we that have derived about the string theory from the counting of the sYTs using these walks. In particular, the following picture emerges:

1. Every state in the Hilbert space is assigned to two independent Motzkin walks — one corresponding to the sYT for the boxes, and the other for the anti-boxes. Because the symmetry group is $PU(N)$, these two Motzkin walks have the same number of steps. In the gauged model, we may think of these two Motzkin walks as corresponding to the state in the irrep of the $\mathsf{S}_{\Bbbk}$ that permutes the adjoints.

2. Given a state with $\Bbbk$ steps, each of the two Motzkin walkers can explore any number of the available dimensions. However, in strict analogy with what happens with the

---

[33]Rather unhelpfully, there are two separate notions of a Motzkin walk with colour degrees of freedom that appear in the literature. Unfortunately, despite sharing the same name, these two generalizations of the Motzkin walk have very different properties and it is therefore important that a distinction is made for our discussion. We will not describe the other generalization of the walk or the differences between the two here, but only provide references for the interested reader. The generalization of the Motzkin walk that we use is defined in [61]. The other generalisation of the coloured Motzkin walk appears in [62–64].

[34]A slightly smaller set of walks can be associated to sYTs with $2s$ rows. As elsewhere in this paper, we ignore all even-odd effects.

sYTs, the dominant contribution to the entropy arises from the walks that explore $\mathcal{O}(\sqrt{\mathbb{k}})$ dimensions.

3. If we now calculate the entropy of all the doubled Motzkin walks traversing $\mathbb{k}$ steps, we see that this is exactly equal to the renormalized entropy (139), i.e.

$$S_{\mathrm{ren},\mathbb{k}} = 2\,\mathbb{k}\log\sqrt{\mathbb{k}}. \tag{153}$$

In other words, the Motzkin walks appear to directly compute quantities associated with the fixed-charge partition function in the gauged model, $Z_{\mathrm{ren},\mathbb{k}}$.

Bolstered by this last observation, it is extremely tempting to 'forget' about the sYTs entirely and attempt to draw a deeper and more direct connection between these Motzkin walks and the string theory on the cigar background. This idea is not new — beginning with the work of Horowitz and Polchinski [65], a common motif in various investigations of string theory near the Hagedorn transition have involved drawing parallels with various random walk models (see [66–68] and references therein). The Horowitz-Polchinski solution (see [69] for a recent review) describes a self-gravitating gas of strings that then condense into a single stretched string near the Hagedorn transition. The Euclidean picture of this computation involves a condensate that winds around the thermal circle that grows massless at the Hagedorn temperature — in a first quantized description, the path integral for this field is just a sum over random walks.

A similar cartoon emerges while studying the Euclidean cigar. Near the Hagedorn transition there is a winding condensate that is concentrated at the tip of the geometry. More precisely, the profile of this winding mode is described by the Nambu-Goto action of a string stretching from the tip of the cigar up to a particular radial distance. As the work of [51] emphasize, it is this condensate that carries the entropy of this system. We suspect that the Motzkin walks can provide a direct Hamiltonian realization of this entropy.

If this picture is true, it has two neat consequences. First, [70] draws a connection between the cigar random walks and 'string bits'. These are fundamental point-like objects first introduced in [71] in order to make the causality and stability of string theory manifest — strings are then regarded as composite 'multi-bit' objects with a particular linear arrangement [72]. This would allows us to interpret each step of the Motzkin walk as an individual bit. Secondly, the picture that we have drawn describes a repackaging of the dynamics of a gas of strings (akin to dealing with multiple adjoints) into Motzkin walks (which, like the sYTs are simply irreps with the correct number of boxes). In other words, Schur-Weyl duality reorganises the physics of the system in a manner very reminiscent of ER=EPR duality!

Of course, there is much work to be done before we can make this picture precise. To begin with, here are some simple questions:

1. Why are there two Motzkin walks for each state?

2. What does the restriction of the walk to the positive quadrant of the $\mathbb{R}^{s+1}$ mean from the perspective of the string theory?

3. What do the dimensions (colours) correspond to in the string theory?

We don't have satisfactory answers to these questions yet — we leave them here as launching pads for future exploration.

## Acknowledgments

We thank Matt Blacker, Nick Dorey, Abhijit Gadde, Adwait Gaikwad, Chris Herzog, Sean Hartnoll, Vladimir Kazakov, Igor Klebanov, Juan Maldacena, Raghu Mahajan, Shiraz Minwalla, Prahar Mitra, Onkar Parrikar, Steve Shenker, Eva Silverstein, Sandip P. Trivedi, Arkady Tseytlin, Aron Wall, Toby Wiseman, Edward Witten, Zhenbin Yang, and Shunyu Yao for discussions. We especially thank Juan Maldacena for sharing some unpublished notes. We thank Panos Betzios, Juan Maldacena and Olga Papadoulaki for comments on a draft.

**Funding information** This work has been partially supported by STFC consolidated grant ST/T000694/1. RMS is supported by the Isaac Newton Trust grant "Quantum Cosmology and Emergent Time" and the (United States) Air Force Office of Scientific Research (AFOSR) grant "Tensor Networks and Holographic Spacetime". KR is supported by the Rhodes Trust via a Rhodes Scholarship. AF is partially supported by the NSF GRFP under grant no. DGE-165-6518. KR would like to thank Cambridge University and DAMTP for their gracious hospitality while a part of this work was being completed.

## A  Entropy count with $\Bbbk \gg N^2$ boxes

In this case, we have not been able to calculate the value of the energy; however, the $PU(N)$ symmetry guarantees a large degenerate subspace and so we might simply count its dimension as an exercise.

Focusing on just the boxes for now, a typical Young diagram $r$ has $N/2$ rows with row lengths

$$\lambda_\alpha = \frac{2\Bbbk}{N} + \delta\lambda_\alpha, \quad \frac{2\Bbbk}{N} \gg N, \quad \sum_\alpha \delta\lambda_\alpha = 0. \tag{A.1}$$

Let us calculate the dimension of this irrep. The formula for the dimension is [49]

$$\dim \mathscr{H}_r = \frac{1}{\prod_{\alpha, a_\alpha \in 1\dots\lambda_\alpha} hl(\alpha, a_\alpha)} \prod_\alpha \frac{(N - (\alpha - 1) + \lambda_\alpha)!}{(N - (\alpha - 1))!}, \tag{A.2}$$

where $hl(\alpha, a)$ is the hook length of the $a^{th}$ box on the $\alpha^{th}$ row.

Because the length of the tableau is much bigger than its height, we can approximate the hook length by ignoring the boxes below any given box,

$$hl(\alpha, a) \approx \lambda_\alpha - a, \quad \Rightarrow \quad \prod hl(\alpha, a) \approx \prod_\alpha \lambda_\alpha!. \tag{A.3}$$

So, the dimension is

$$\dim \mathscr{H}_{r_1} \approx \prod \binom{N - \alpha + \lambda_\alpha}{\lambda_\alpha} \approx \left(\frac{\sqrt{\Bbbk}}{N}\right)^{\frac{3}{4}N^2} \dim \mathscr{H}_{r_1} \approx \prod \binom{N - \alpha + \lambda_\alpha}{\lambda_\alpha} \approx \left(\frac{\sqrt{\Bbbk}}{N}\right)^{\frac{3}{4}N^2}, \tag{A.4}$$

To include the anti-boxes, we merely need to square this. So, we find that

$$\dim \mathscr{H}_r = \dim \mathscr{H}_{r_1 \otimes \bar{r}_2} \approx \left(\frac{\sqrt{\Bbbk}}{N}\right)^{\frac{3}{2}N^2}. \tag{A.5}$$

So, our conjecture for the entropy in this phase is

$$S_{\Bbbk} = \frac{3}{2} N^2 \log \frac{\sqrt{\Bbbk}}{N}. \tag{A.6}$$

Note that the growth slows down from linear to logarithmic in $\Bbbk$.

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
