# Peer review of "Boundary Description of Microstates of the Two-Dimensional Black Hole"

_SciPost Physics, doi:SciPost Phys. 16, 020 (2024)_

## Round 1 · Author Response

We thank the reviewers for their detailed and thoughtful reports.
They have helped us improve our presentation.
Below, we answer the many good points raised by both reviewers and point to changes made in the draft.
\section{Report 1}
\subsection{Major Points}
\begin{enumerate}
\item We are unsure which part of the analysis the reviewer found too qualitative.
We have endeavoured in writing the paper to keep the qualitative arguments in the discussion section, even at the cost of some non-linearity in the presentation.
If the reviewer could clarify this point, we would be happy to edit the draft suitably.
\item Regarding the interpretation as a black hole, we thank the reviewer for catching the inadequacy of our discussion of this point.
We have added a caveat before section 2.1.
\item The confusion at the beginning of section 3 is due to a certain non-linearity in our presentation.
We have rephrased those paragraphs for clarity.
\item It is indeed true that the wavefunction we work out is related to \emph{tips} of the long strings in the interpretation of [16].
We have discussed this point in section 6.3, in a qualitative way.
The large entropy found by Kazakov and Tseytlin arises from the uncertainty of the placement of the string \emph{end-points}, and that is why it gets renormalised away in the double-scaling limit.
%The answer to this point is that for $\mathbb{k} \gg \log N$, the adjoints are energetically pushed into the double scaling limit. For $\mathbb{k} \ll \log N$, we assume that the adjoints are supported in the double scaling limit, and find that this state is metastable. I don't know where we can make this point clearer in the draft.
\item It is true that $\lambda_{\nu+1}$ is related to the sine-Liouville coupling constant.
We give a numerological discussion in section 6.1, and some follow-up calculations not included in this work supports this intuition.
The question of whether singlets can tunnel across the hole has not been addressed in this work, but we can give a concise account of what a subset of the authors have learnt in follow-up work.
The size of the gap scales with $\lambda_{\nu+1} - \lambda_{\nu} \sim \mathbb{k}^{1/4}$, and so the tunnelling probability vanishes exponentially at large $\mathbb{k}$.
The limit in which sine-Liouville theory emerges is $N \gg \mathbb{k} \gg \mu$, and so this tunnelling probability vanishes and does not seem to be visible in the sine-Liouville theory.
It is unclear that the black hole absorbs singlets in the string theory; for example, the S-matrices in sine-Liouville theory don't seem to have poles at imaginary energy.
Non-singlets/string tips, however, can jump across the gap in the relevant limit, but we have not yet pinned down a signature of this in sine-Liouville.
\item Our wavefunction does also carry spin degrees of freedom - these are encoded in the labels of the Young tableaux, see e.g. eqn (3.4).
\item The reverse process, of absorbing non-singlets, should certainly be taken into account.
We have not done the relevant calculations.
This question is tied into the larger question of why these states dominate the fixed-charge canonical ensemble in the double-scaling limit.
We have argued that this state dominates when $\mathbb{k} > \log N$, but this is not the double-scaling limit.
In the double-scaling limit, the lowest energy state is one where all the string tips are infinitely far away.
We have added some discussion of this just before section 4.3.1.
\end{enumerate}
\section{Minor Points}
\begin{enumerate}
\item The reviewer is correct about the scaling of the mass, but we are uncertain about how it contradicts (2.2).
\item Regarding (2.6), this is a T-dualised version. See references [11,12] for details.
\item Certainly, the entanglement on the EPR side is carried by the long strings, but these are already included in the sine-Liouville term.
We have added a clarification where we mention this point.
\item We have clarified that (2.16) only holds for $R<2$, with thanks for pointing out this omission.
\item It is in fact the same notation. [16] followed [38] here, and we clarify that we are also following [38].
\item (4.16) and (4.17) are already in different paragraphs. Perhaps there is a typo in the report?
\item We have added a citation to a review on the spin-Calogero model.
\end{enumerate}
\section{Report 2} \label{sec:rep2}
\begin{enumerate}
\item At finite $N$, this hole-in-the-world state dominates the fixed-charge canonical partition function, as we argue in section 4.3.
We have added some more discussion of the double-scaling limit just before section 4.3.1.
\item 4.55 is a numerological observation.
The remainder of section 4.2 argues that these values arise from a minimisation of energy.
It is important here that there is no physics input, since any such would invalidate the minimisation.
The physics of the answer is discussed in section 6.1.
We agree that the phrasing around 4.55 was confusing, and we have modified it to clarify this.
\end{enumerate}
They have helped us improve our presentation.
Below, we answer the many good points raised by both reviewers and point to changes made in the draft.
\section{Report 1}
\subsection{Major Points}
\begin{enumerate}
\item We are unsure which part of the analysis the reviewer found too qualitative.
We have endeavoured in writing the paper to keep the qualitative arguments in the discussion section, even at the cost of some non-linearity in the presentation.
If the reviewer could clarify this point, we would be happy to edit the draft suitably.
\item Regarding the interpretation as a black hole, we thank the reviewer for catching the inadequacy of our discussion of this point.
We have added a caveat before section 2.1.
\item The confusion at the beginning of section 3 is due to a certain non-linearity in our presentation.
We have rephrased those paragraphs for clarity.
\item It is indeed true that the wavefunction we work out is related to \emph{tips} of the long strings in the interpretation of [16].
We have discussed this point in section 6.3, in a qualitative way.
The large entropy found by Kazakov and Tseytlin arises from the uncertainty of the placement of the string \emph{end-points}, and that is why it gets renormalised away in the double-scaling limit.
%The answer to this point is that for $\mathbb{k} \gg \log N$, the adjoints are energetically pushed into the double scaling limit. For $\mathbb{k} \ll \log N$, we assume that the adjoints are supported in the double scaling limit, and find that this state is metastable. I don't know where we can make this point clearer in the draft.
\item It is true that $\lambda_{\nu+1}$ is related to the sine-Liouville coupling constant.
We give a numerological discussion in section 6.1, and some follow-up calculations not included in this work supports this intuition.
The question of whether singlets can tunnel across the hole has not been addressed in this work, but we can give a concise account of what a subset of the authors have learnt in follow-up work.
The size of the gap scales with $\lambda_{\nu+1} - \lambda_{\nu} \sim \mathbb{k}^{1/4}$, and so the tunnelling probability vanishes exponentially at large $\mathbb{k}$.
The limit in which sine-Liouville theory emerges is $N \gg \mathbb{k} \gg \mu$, and so this tunnelling probability vanishes and does not seem to be visible in the sine-Liouville theory.
It is unclear that the black hole absorbs singlets in the string theory; for example, the S-matrices in sine-Liouville theory don't seem to have poles at imaginary energy.
Non-singlets/string tips, however, can jump across the gap in the relevant limit, but we have not yet pinned down a signature of this in sine-Liouville.
\item Our wavefunction does also carry spin degrees of freedom - these are encoded in the labels of the Young tableaux, see e.g. eqn (3.4).
\item The reverse process, of absorbing non-singlets, should certainly be taken into account.
We have not done the relevant calculations.
This question is tied into the larger question of why these states dominate the fixed-charge canonical ensemble in the double-scaling limit.
We have argued that this state dominates when $\mathbb{k} > \log N$, but this is not the double-scaling limit.
In the double-scaling limit, the lowest energy state is one where all the string tips are infinitely far away.
We have added some discussion of this just before section 4.3.1.
\end{enumerate}
\section{Minor Points}
\begin{enumerate}
\item The reviewer is correct about the scaling of the mass, but we are uncertain about how it contradicts (2.2).
\item Regarding (2.6), this is a T-dualised version. See references [11,12] for details.
\item Certainly, the entanglement on the EPR side is carried by the long strings, but these are already included in the sine-Liouville term.
We have added a clarification where we mention this point.
\item We have clarified that (2.16) only holds for $R<2$, with thanks for pointing out this omission.
\item It is in fact the same notation. [16] followed [38] here, and we clarify that we are also following [38].
\item (4.16) and (4.17) are already in different paragraphs. Perhaps there is a typo in the report?
\item We have added a citation to a review on the spin-Calogero model.
\end{enumerate}
\section{Report 2} \label{sec:rep2}
\begin{enumerate}
\item At finite $N$, this hole-in-the-world state dominates the fixed-charge canonical partition function, as we argue in section 4.3.
We have added some more discussion of the double-scaling limit just before section 4.3.1.
\item 4.55 is a numerological observation.
The remainder of section 4.2 argues that these values arise from a minimisation of energy.
It is important here that there is no physics input, since any such would invalidate the minimisation.
The physics of the answer is discussed in section 6.1.
We agree that the phrasing around 4.55 was confusing, and we have modified it to clarify this.
\end{enumerate}

---

## Round 1 · List of Changes

- Added para before section 2.1.
- Added a comment about the relation to ER=EPR after eqn 2.9.
- Clarified discussion before section 3.1.
- Added reference on spin-Calogero models after eqn 4.19.
- Rewrote beginning of section 4.2.3 to make it clearer.
- Wrote an extended explanation of an open question in the last paragraph before section 4.3.1.

---

## Editorial Decision

published